 **eLIFE**

# A druggable secretory protein maturase of Toxoplasma essential for invasion and egress

**Sunil Kumar Dogga[1], Budhaditya Mukherjee[1], Damien Jacot[1], Tobias Kockmann[2], Luca Molino[1], Pierre-Mehdi Hammoudi[1], Ruben C Hartkoorn[1,3], Adrian B Hehl[4], Dominique Soldati-Favre[1]***

[1]Department of Microbiology and Molecular Medicine, University of Geneva, Geneva, Switzerland; [2]Functional Genomics Center Zurich, ETH Zurich/University of Zurich, Zurich, Switzerland; [3]Chemical Biology of Antibiotics, Center for Infection and Immunity, Inserm U1019, CNRS UMR8204, Institut Pasteur de Lille, Lille, France; [4]Institute of Parasitology, University of Zurich, Zurich, Switzerland

**Abstract** Micronemes and rhoptries are specialized secretory organelles that deploy their contents at the apical tip of apicomplexan parasites in a regulated manner. The secretory proteins participate in motility, invasion, and egress and are subjected to proteolytic maturation prior to organellar storage and discharge. Here we establish that *Toxoplasma gondii* aspartyl protease 3 (ASP3) resides in the endosomal-like compartment and is crucially associated to rhoptry discharge during invasion and to host cell plasma membrane lysis during egress. A comparison of the N-terminome, by terminal amine isotopic labelling of substrates between wild type and ASP3 depleted parasites identified microneme and rhoptry proteins as repertoire of ASP3 substrates. The role of ASP3 as a maturase for previously described and newly identified secretory proteins is confirmed *in vivo* and *in vitro*. An antimalarial compound based on a hydroxyethylamine scaffold interrupts the lytic cycle of *T. gondii* at submicromolar concentration by targeting ASP3.
DOI: https://doi.org/10.7554/eLife.27480.001

**\*For correspondence:**
Dominique.Soldati-Favre@unige.ch

## Introduction

Members of the Apicomplexa phylum include obligate intracellular parasites that are responsible for severe diseases in humans and farm animals. *Toxoplasma gondii* infection can be lethal for immuno-compromised individuals, while infection in pregnant women can lead to birth defects or miscar-riage. *P. falciparum* is responsible for the most life-threatening form of malaria and globally is one of the top ten causes of death with half the world's population currently at risk. Additionally, other Api-complexans such as *Cryptosporidium*, *Eimeria*, *Theileria* and *Babesia* spp. infect farm animals and are responsible for considerable economic losses. No efficient vaccines or eradicating drug treat-ments are available against this important group of pathogens.

Invasion and egress are two key steps in the lytic cycle of Apicomplexa as they go through com-plex life stage development in one or multiple hosts (*Dubey et al., 1998*). The sequential and regu-lated discharge of protein factors from two apical secretory organelles called micronemes and rhoptries is essential for parasite survival and for dissemination of the infection in the host (*Carruthers and Sibley, 1997*). The content of these two types of organelles includes notably adhe-sins, perforins, proteases, lipases and kinases that are critical for motility, host attachment, formation of the parasitophorous vacuole membrane (PVM) during invasion and for lysis of the PVM and host plasma membrane upon egress (*Kemp et al., 2013*; *Roiko and Carruthers, 2009*). The microneme (MICs), rhoptry neck (RONs) and rhoptry bulb (ROPs) proteins are produced late during the parasite

cell cycle and undergo extensive proteolytic maturation in an endosomal-like compartment (ELC) along their trafficking from the Golgi to their respective target organelles (*Nishi et al., 2008*).

MICs typically exist as complexes composed of adhesive and transmembrane domain-containing proteins that are discharged onto the parasite surface at the apical tip. Some of these complexes engage with host receptors and translocate to the posterior end of the parasite, hence mediating gliding motility and invasion. The characterized complexes to date include MIC2-M2AP (MIC2-associated protein) involved in gliding motility (*Huynh and Carruthers, 2006*; *Huynh et al., 2003*), MIC6-MIC1-MIC4 contributing to parasite attachment and invasion (*Reiss et al., 2001*), and MIC3-MIC8 associated with rhoptry secretion (*Cérède et al., 2002*; *Kessler et al., 2008*). The majority of the MICs undergo extensive pre- and post-exocytosis proteolytic processing. Intracellular processing occurs post-Golgi and prior to the storage in the micronemes. The MIC pro-domains have been implicated in facilitating traffic to microneme organelles. A propeptide deletion mutant of M2AP is retained in the post-Golgi region, hampering its trafficking to the micronemes (*Harper et al., 2006*). While the cleavage mutant of M2AP shows normal trafficking to micronemes, the removal of the pro-domain appears to be important for MIC2-M2AP complex formation and secretion (*Harper et al., 2006*). The first of the three EGF-like domains of MIC6 is cleaved off, but this event does not impact on the MIC1-MIC4-MIC6 complex formation and trafficking (*Reiss et al., 2001*). Similarly, MIC3, MIC5 and M2AP are processed prior to reaching the micronemes (*Brydges et al., 2000*; *Cérède et al., 2002*; *Rabenau et al., 2001*). Once on the parasite surface, the transmembrane MICs, are shed by the microneme protein protease 1 (MPP1) activity that has been attributed to rhomboid protease dependent intramembrane proteolysis (*Rugarabamu et al., 2015*; *Shen et al., 2014b*). In addition, the MPP2 activity mediated by the action of subtilisin protease SUB1 (SUB1) also participates in post-exocytosis processing of the MICs (*Lagal et al., 2010*; *Saouros et al., 2012*). The protease responsible for the pre-exocytosis processing of these secreted proteins has not been identified to date, although the presence of cathepsin L (CPL) is required for optimal processing (*Parussini et al., 2010*).

Following attachment and apical reorientation of the parasite towards the host cell, the rhoptries' membranous material and proteins are directly discharged into the host cell cytosol during parasite entry (*Carruthers and Boothroyd, 2007*). A subset of proteins at the neck of the rhoptries, termed rhoptry neck proteins (RONs) form a complex (RON2-4-5-8) that is anchored into the host cell membrane and associated with the microneme protein AMA1 on the parasite surface (*Alexander et al., 2005*; *Besteiro et al., 2009*; *Lebrun et al., 2005*). This AMA1-RONs complex plays a central role in moving junction (MJ) formation and host cell entry (*Bargieri et al., 2013*; *Lamarque et al., 2014*; *Mital et al., 2005*). The MJ presumably serves as a support for the propulsion of the parasite into the host cell powered by the actomyosin system and the connector GAC (*Jacot et al., 2016*). The rhoptry bulb proteins (ROPs), on the other hand, act as effectors and subvert host cellular functions by either interacting with the PVM to protect the parasites against host defense and clearance mechanisms or trafficking to the host nucleus to reprogram gene expression (*Bradley et al., 2005*; *Håkansson et al., 2001*). Many ROPs contain kinase and pseudokinase domains (*El Hajj et al., 2006*; *Peixoto et al., 2010*) and act as key virulence factors such as ROP18, which along with ROP5 promotes parasite survival in mice by inactivating immunity-related GTPase (IRGs) by phosphorylation (*Niedelman et al., 2012*) and ROP16, which interacts with the host cell STAT signaling cascade, phosphorylating STAT3/STAT6 and suppressing TLR signaling (*Ong et al., 2010*; *Saeij et al., 2007*; *Yamamoto et al., 2009*).

Most of the rhoptry proteins are synthesized with an N-terminal pro-domain subsequently cleaved in a post-Golgi compartment. The ROPs, in particular, contain a putative cleavage site with the consensus sequence SφXE (φ is hydrophobic, X is any amino acid), and presumably processed by the rhoptry subtilisin SUB2 within the immature rhoptries of nascent daughter parasites (*Miller et al., 2003*). The cleavage site has been experimentally confirmed only for ROP1, ROP13 and SUB2 (*Bradley and Boothroyd, 1999*; *Miller et al., 2003*; *Soldati et al., 1998*; *Turetzky et al., 2010*). Though the pro-domain of ROP1 is shown to be sufficient for rhoptry targeting, the removal of the pro-domain does not appear to affect its targeting to the organelles (*Bradley et al., 2002*; *Miller et al., 2003*; *Turetzky et al., 2010*) and few of the ROPs, e.g. ROP5, are not processed during their trafficking. RONs are also extensively processed along the secretory pathway, and have SφXE consensus motifs within their sequences (*Besteiro et al., 2009*), however, the proteases implicated in their maturation are not known.

*T. gondii* encodes 7 aspartyl proteases (ASPs), with only ASP1, ASP3 and ASP5 being significantly expressed in the tachyzoite stage. Among these, ASP1 is coccidian-specific and dispensable in tissue culture as well as in the mouse model of infection (*Polonais et al., 2011*). The Golgi-associated aspartyl protease ASP5 has recently been associated to the establishment of the intravacuolar network and the trafficking of effectors to the PVM and beyond (*Coffey et al., 2015*; *Curt-Varesano et al., 2016*; *Hammoudi et al., 2015*).

Herein, we report the functional dissection of ASP3 and demonstrate its essential nature for invasion and egress from the host cells. We exploited the power of a proteomics pipeline based on the novel terminal amine isotopic labelling of substrates (TAILS) technology (*Kleifeld et al., 2010*; *Kleifeld et al., 2011*) to compare the N-terminome of wild type and ASP3-depleted parasites. The TAILS analysis identified the repertoire of ASP3 substrates and revealed that ASP3 acts as maturase for ROPs, RONs and MICs along their trafficking route from Golgi to their respective organelles. We validated a series of known as well as novel ASP3 substrates both in vivo and in vitro toward a molecular understanding of the dramatic phenotype of ASP3 depletion in invasion and egress. Importantly, a hydroxyethylamine scaffold based compound, 49c, disrupts the *T. gondii* lytic cycle by inhibiting ASP3 at submicromolar concentration.

## Results

### ASP3 is a post-Golgi resident aspartyl protease essential for invasion and egress

In the phylum of Apicomplexa, aspartyl proteases cluster into 6 distinct clades (*Figure 1—figure supplement 1*). Members of this class of proteases exhibit two catalytic aspartic acid residues (conserved motifs DTG or DTG/DSG), a polyproline loop and a flap predicted to close over the active site, thereby influencing substrate binding. (*Figure 1—figure supplement 2*). ASP3 groups together with two phylogenetically related genes in the subgroup Haemosporidia, that correspond to Plasmepsin IX and X (PMIX and PMX) in the malaria parasites. To characterize ASP3, we first tagged C-terminally its endogenous locus with 3xty tag (*Figure 1—figure supplement 3*). Western blot analysis (WB) of the tagged strain (ASP3ty) revealed two major forms, a band at ~70 kDa corresponding to the predicted full-length protein, and a 50 kDa processed form of the enzyme (*Figure 1A*). Ectopically expressed ASP3 was previously localized to a post-Golgi compartment (*Shea et al., 2007*), which was confirmed in the ASP3ty strain (*Figure 1B*). Indirect immunofluorescence analysis (IFA) using markers for various compartments of the secretory pathway (GRASP – cis-Golgi, DrpB – post-Golgi, proM2AP – ELC) revealed that ectopically expressed ASP3 localizes in the vicinity of the ELC and beyond (*Breinich et al., 2009*; *Harper et al., 2006*). Since ASP3 did not co-localize perfectly with any of these markers, we conclude that the enzyme is likely to be present in more than one subcompartment (*Figure 1B–C*).

*ASP3* was refractory to gene excision (data not shown) and in consequence, we generated a conditional knockdown of *ASP3* by endogenous promoter swap with a tet-repressible promoter. The genotypes of the resulting three independent strains (ASP3-iKD, ASP3ty-iKD and ASP3myc-iKD) were confirmed by genomic PCR analysis (*Figure 1—figure supplements 3* and *4*). The tightness of anhydrotetracycline (ATc)-dependent expression in ASP3ty-iKD and ASP3myc-iKD parasites was assessed by WB (*Figure 1A* and *Figure 1—figure supplement 6A*). Plaque assays performed on the ASP3myc-iKD parasites (*Figure 1D*) incubated with or without ATc for 7 days exhibited no plaque formation in absence of ASP3 indicating a severe defect in one or several steps of the lytic cycle and the same phenotype was observed with ASP3ty-iKD (data not shown). Complementation of ASP3myc-iKD with a second copy of ASP3ty fully rescued the lethality in the presence of ATc, while a catalytically dead mutant of ASP3 (asp3ty-D299A) failed to rescue (*Figure 1D* and *Figure 1—figure supplement 5*). Interestingly, asp3ty-D299A is only weakly processed in the absence of ATc whereas the processing completely disappeared upon ASP3 depletion, suggesting an autocatalytic maturation process (*Figure 1—figure supplement 6A–B*).

A deeper investigation of each steps of the lytic cycle revealed that ASP3 depletion did not affect parasite intracellular growth, parasite attachment or gliding motility whereas invasion and induced egress were severely impaired (*Figure 1E–I*). The mitochondrion and apicoplast appear morphologically normal in absence of ASP3 (*Figure 1—figure supplement 6C*). Relevantly, ASP3-depleted

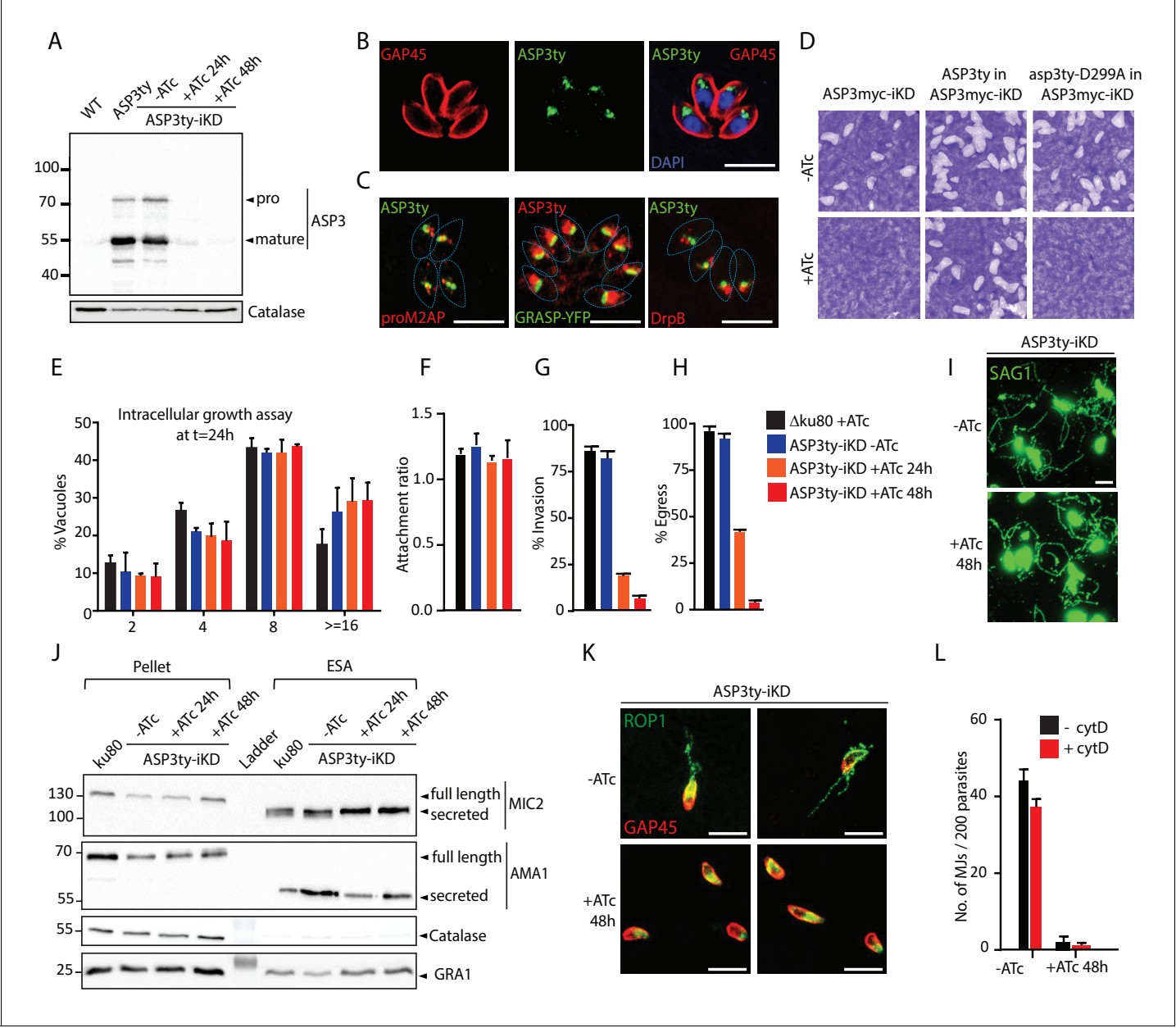

**Figure 1.** ASP3 is present in the secretory pathway and is essential for invasion and egress. (**A**) Ty-tagged endogenous ASP3 (ASP3ty) expression and tight regulation of ASP3ty tet-inducible knockdown (ASP3ty-iKD -/+ATc) assessed by western blot. Catalase was used as loading control. (**B**) ASP3 localizes to a post-Golgi compartment. (**C**) Ty-tagged ASP3 expressed from a tubulin promoter partially colocalized with GRASP-YFP, proM2AP and DrpB. (**D**) ASP3 depletion results in drastic impairment of the lytic cycle, as assessed by plaque formation after 7 days, in the presence of ATc. Complementation with a second WT copy of ASP3 (ASP3ty) fully restored plaque formation whereas complementation with a catalytically dead mutant (asp3ty-D299A) could not. ASP3 depleted parasites (24 or 48 hr +ATc) showed no defect in E) intracellular replication and in F) attachment, but showed a strong defect in G) invasion and H) egress compared to control RHΔku80 (48 hr +ATc) or non-treated ASP3ty-iKD parasites. Data are presented as mean ±standard deviation (SD) from 3 independent experiments. (**I**) ASP3 depleted parasite (48 hr +ATc) showed no defect in gliding motility. Gliding trails are detected using anti-SAG1 antibodies. (**J**) Western blot analyses of excreted-secreted antigens (ESA) and pellets from ASP3ty-iKD parasites, +/−ATc, after stimulation with 2% Ethanol (EtOH) showed normal secretion of the microneme proteins MIC2 and AMA1. Full length and secreted fragments are showed with arrowheads. Catalase was used as cytosolic control and GRA1 for constitutive secretion. (**K**) ASP3 depletion resulted in a complete absence of rhoptry content secretion as assessed by evacuole formation by probing against the rhoptry protein ROP1. (**L**) The formation of the MJ was assessed by the specific labelling of secreted RON4 and showed to be dependent on the presence of ASP3. This was independent of cytochalasin D treatment. Data are represented as mean ±standard deviation of three independent biological replicate experiments. All scale bars throughout represent 8 µm.

DOI: https://doi.org/10.7554/eLife.27480.002

*Figure 1 continued on next page*

*Figure 1 continued*

The following source data and figure supplements are available for figure 1:

**Source data 1.** Source data of the triplicate experiments done on the ASP tet-inducible knockdown strains (-ATc, +ATc 24 hr and +ATc 48 hr) for Intracellular growth assay (E), Host cell attachment assay (F), Invasion assay (G), Egress assay (H) and quantification of MJs (L).
DOI: https://doi.org/10.7554/eLife.27480.009
**Figure supplement 1.** Phylogenetic clustering of apicomplexan aspartyl proteases.
DOI: https://doi.org/10.7554/eLife.27480.003
**Figure supplement 2.** *T. gondii* aspartyl proteases and its *P. falciparum* homologs.
DOI: https://doi.org/10.7554/eLife.27480.004
**Figure supplement 3.** Strategy for epitope tagging at the endogenous locus.
DOI: https://doi.org/10.7554/eLife.27480.005
**Figure supplement 4.** Generation of the tet-inducible ASP3 strains.
DOI: https://doi.org/10.7554/eLife.27480.006
**Figure supplement 5.** Complementation in tet-inducible ASP3 strain with WT ASP3ty and the catalytically dead mutant asp3ty-D299A.
DOI: https://doi.org/10.7554/eLife.27480.007
**Figure supplement 6.** Phenotyping of ASP3 depleted parasites.
DOI: https://doi.org/10.7554/eLife.27480.008

parasites showed delayed exit from the host cells upon both natural and induced egress by live microscopy. Motile parasites were trapped in spherical detached host cells and eventually rupturing the host PM (*Figure 1—figure supplement 6D*, *Videos 1–3*). This egress defect is reminiscent of the reported phenotype in parasites lacking the microneme perforin, PLP1 (*Kafsack et al., 2009*).

Freshly egressed ASP3ty-iKD and WT parasites, pre-treated or not with ATc, were stimulated with 2% ethanol to induce micronemes secretion. The resulting excreted secreted antigens (ESA) fractions were analyzed by WB and revealed no significant alteration in microneme secretion. The dense granule protein, GRA1, and the cytosolic catalase, CAT, served as controls for parasite viability and parasite lysis, respectively (*Figure 1J*).

Rhoptry discharge can be assessed by analysis of empty vacuoles (evacuoles) formation in presence of cytochalasin D (cytD), an actin polymerization inhibitor known to block motility and invasion without impairing microneme and rhoptry secretion (*Håkansson et al., 2001*; *Kessler et al., 2008*). The injection of the rhoptry content in host cells is detectable by IFA using antibodies recognizing the rhoptry protein ROP1. Strikingly, parasites depleted in ASP3 failed to produce evacuoles (*Figure 1K*). The secretion of RONs can be monitored by assessing MJ formation by IFA on 0.1%

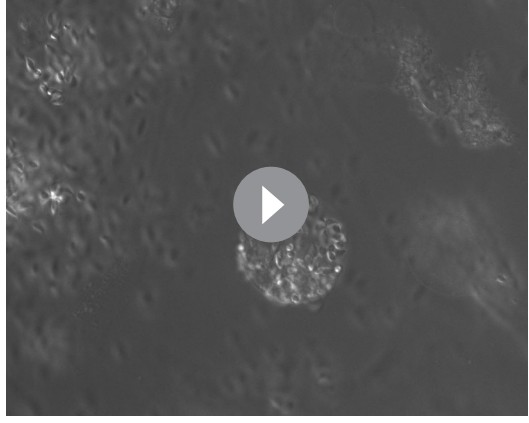

**Video 1.** ASP3 depleted parasites display impaired egress. During egress from host cell, ASP3 depleted parasites were often found trapped in floating spherical membranous structures. The parasites are very motile within these structures and attempt to mechanically rupture the membranes with most eventually escaping.
DOI: https://doi.org/10.7554/eLife.27480.010

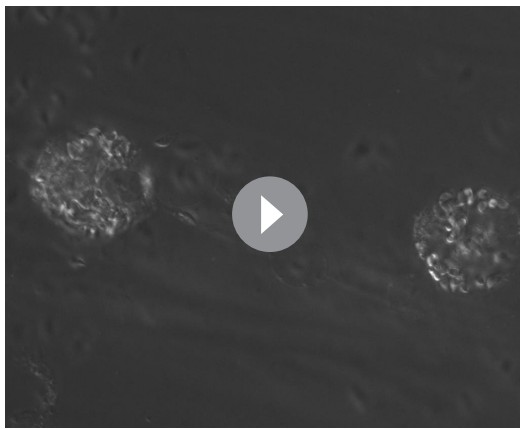

**Video 2.** ASP3 depleted parasites display impaired egress. During egress from host cell, ASP3 depleted parasites were often found trapped in floating spherical membranous structures.
DOI: https://doi.org/10.7554/eLife.27480.011

saponin permeabilized parasites. In ASP3-depleted parasites, RON4 was absent at the junction in more than 90% of parasites, with or without prior cytD treatment (*Figure 1L* and *Figure 1—figure supplement 6E*). Collectively, these results reveal that ASP3 is necessary for the lysis of the host PM and for rhoptry content discharge, explaining the severe blocks in egress and invasion respectively.

## Identification of ASP3 substrates using N-terminome analysis

The phenotypic consequences of ASP3 depletion suggest that this protease contributes to the processing of proteins implicated in rhoptry discharge and host PM lysis. Given its localization in the vicinity of the ELC, we hypothesized a plausible role as maturase for MICs, ROPs and RONs, known to undergo maturation along the secretory pathway, prior to accumulation in their respective secretory organelles. To determine the *in vivo* repertoire of ASP3 substrates and their precise cleavage sites, we quantitatively compared the N-terminome of ASP3ty-iKD parasites +ATc/-ATc using Terminal Amine Isotopic Labelling of Substrates (TAILS) (*Figure 2—figure supplement 1*) (*Kleifeld et al., 2010*; *Kleifeld et al., 2011*). The TAILS datasets yielded 32594 peptide spectrum match (PSM), which could be assigned to 9169 *T. gondii*-specific peptide groups representing 1529 proteins contained in ToxoDB. Among the identified peptides, we found 1487 N-terminal peptides (~16%) supported by 8103 PSMs. 615 of these terminal peptides were acetylated (supported by 2597 PSMs) and therefore represent natural N-termini. The remaining 872 peptide groups were labelled with TMT on the alpha amine group (*Figure 2A*). As expected, the total proteome (pre-TAILS data, 1236 annotated proteins) overlapped with the proteins identified by the TAILS dataset (887 proteins), but enrichment for N-terminal peptides also identified 293 additional proteins (*Figure 2B*). Normalized log2 abundance ratios (+ATc/-ATc) were calculated for each peptide group to identify ASP3-dependent cleavages.

Among the 872 labelled peptide groups, we identified 65 showing either <0.22 or >2 +ATc/-ATc ratios in the ASP3 depleted condition, representing 17 and 29 proteins (41 unique proteins combined), respectively (*Figure 2C*, *Table 1*, *Supplementary files 1* and *3*). 26 of the 41 unique proteins with putative ASP3-dependent processing profiles are annotated as secreted proteins in ToxoDB. A more relaxed +ATc/-ATc ratio threshold of <0.5 expanded this candidate dataset by 42 proteins to 59 (*Supplementary file 2*). Strikingly, concordant with the egress and invasion phenotypes of ASP3 depletion, the majority of proteins are annotated as MICs, RONs and ROPs, some of which were previously shown to be processed post-Golgi. (Detailed analysis of the TAILS and the criteria for substrate selection is described in Appendix 1).

## ASP3 is necessary for the processing of a subset of microneme proteins

To validate the ASP3 substrates deduced from the TAILS analysis, we examined previously characterized MICs, known to be processed, by WB in ASP3ty-iKD +/-ATc. In agreement with the TAILS data, M2AP, MIC3 and MIC6 accumulated as unprocessed forms at 24 hr post ATc treatment whereas MIC2, MIC4, CPL and MIC8 are not affected (*Figure 3A–B* and *Figure 3—figure supplement 1A*). However, the organellar targeting of these unprocessed MICs remained unchanged (*Figure 3G* and *Figure 3—figure supplement 1B*). This is evident when using antibodies raised against the prodomain of M2AP, which only stain the ELC in wild type parasites but atypically detect the unprocessed M2AP in the micronemes in ASP3 depleted parasites (*Figure 3G*). The defect in processing is rescued by expressing a second copy of wild type ASP3 but not the catalytically dead mutant asp3ty-D299A, as shown for MIC6 (*Figure 3C*).

MIC8 has previously been reported to be necessary for invasion, selectively participating in rhoptry discharge by an unknown mechanism (*Kessler et al., 2008*). Since MIC8 is not processed pre-exocytosis (*Meissner et al., 2002*) and neither the trafficking nor secretion is unaffected in ASP3ty-iKD parasites (*Figure 3—figure supplement 1B–C*), ASP3-dependent impairment in rhoptry discharge is unlikely to be explained by a defect in MIC8 function.

In contrast to the other MICs, MIC5, which was absent in the TAILS analysis, is processed normally in ASP3 depleted parasites, suggesting the existence of an alternate protease acting as maturase for a subset of MICs (*Figure 3D*). In this context, toxolysin 4 (TLN4) is a metalloproteinase stored in the micronemes, extensively processed prior to secretion and previously reported to participate in invasion and egress (*Laliberté and Carruthers, 2011*). When expressed as a HA-tagged second copy in ASP3ty-iKD, TLN4 exhibited an altered pattern of processing upon ASP3 depletion (*Figure 3E*).

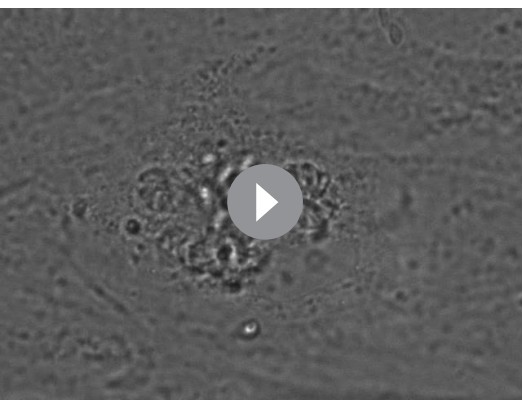

**Video 3.** ASP3 depleted parasites display impaired egress. Motile ASP3 depleted parasites were trapped either in the host cell membrane and/or the PVM. The parasites are very motile within and attempt to mechanically rupture the membranes with most eventually escaping.
DOI: https://doi.org/10.7554/eLife.27480.012

PLP1 is the only microneme protein implicated in egress and for which a peptide is found enriched in the TAILS data set, which is consistent with the reported N-terminal processing of PLP1 prior to exocytosis (*Kafsack et al., 2009*; *Lagal et al., 2010*). Since parasites lacking PLP1 remain trapped into host cells and still surrounded by the PMV, we also assessed the processing of PLP1 in ASP3ty-iKD +/-ATc by C-terminal tagging at the endogenous locus. The processing of PLP1 was difficult to assess by western blot, however, it does not appear to be severely compromised (*Figure 3F*). In consequence, we cannot formally exclude that the ASP3-dependent defect in host plasma membrane lysis is caused by partially impaired PLP1 activity.

## ASP3 indirectly impacts on the post-exocytosis processing of microneme proteins

Following secretion, MICs are proteolytically trimmed, the role of which is less well understood. Although depletion of ASP3 does not impact on microneme secretion (*Figure 1J*), the post-exocytosis processing pattern of some MICs is nonetheless altered with an accumulation of unprocessed forms (*Figure 3H*). In WT parasites, M2AP undergoes multiple cleavages post-exocytosis generating four fragments, MIC2 (100 kDa) is processed at the N-terminus, which removes a 5 kDa fragment and MIC4 (72 kDa) is sequentially cleaved N-terminally to generate 70 kDa and 50 kDa products (*Lagal et al., 2010*). In ASP3 depleted parasites, M2AP secreted products, the 95 kDa secreted form of MIC2 and the 50 kDa form of MIC4 are absent (*Figure 3H–I*). In contrast, the rhomboid–dependent cleavage within the transmembrane domain of MIC2 and AMA1 occurs normally (*Figure 1J* and *Figure 3H*). Consistent with these observations, the rhomboid proteases ROM4 and ROM5 responsible for AMA1 and MIC2 cleavage (*Rugarabamu et al., 2015*; *Shen et al., 2014b*) are known to be constitutively active, whereas SUB1 involved in the post-exocytosis processing of MIC2, MIC4 and M2AP needs to be activated by a maturation process implicating the removal of the SUB1 pro-domain (*Lagal et al., 2010*; *Saouros et al., 2012*). Output data from the TAILS analysis revealed abundant enrichment for SUB1 peptides, indicative of a plausible ASP3 substrate (*Table 1*). IFA analyses performed with either cross-reacting anti-PfSUB1 antibodies or on SUB1ty tagged parasites, revealed no alteration in the trafficking or secretion of SUB1 in the absence of ASP3 (*Figure 3J*). In contrast, the post-exocytosis processing profile of SUB1 was clearly affected (*Figure 3K*). These results suggest a direct or indirect role for ASP3 in SUB1 maturation leading to its activation on the parasite surface.

## ASP3 is necessary for the processing of the ROPs and RONs

ROPs and RONs represented a significant fraction of hits in the TAILS data that prompted us to investigate whether ASP3 could act as protein maturase for the ROPs and RONs. The lysates of ASP3ty-iKD +/−ATc were analyzed by WB and revealed that RONs and ROPs precursors accumulated as non-processed forms in parasites treated for 24 and 48 hr with ATc (*Figure 4A–D*). The targeting of the ROPs/RONs to the rhoptries is not affected by the block of ASP3-dependent processing, (*Figure 4F* and *Figure 4—figure supplement 1A–B*), corroborating previous observations on some of the ROPs (*Bradley et al., 2002*; *Miller et al., 2003*; *Turetzky et al., 2010*). The processing phenotype is complemented by the second copy of wild type ASP3 but not the catalytically dead mutant asp3-D299A, as shown for ROP2-4 (*Figure 4E*). Given the severe defect in invasion caused by ASP3 depletion, we assessed whether unprocessed RONs would be affected in their assembly as a complex. Co-immunoprecipitation experiments confirmed that the RON2 and RON4

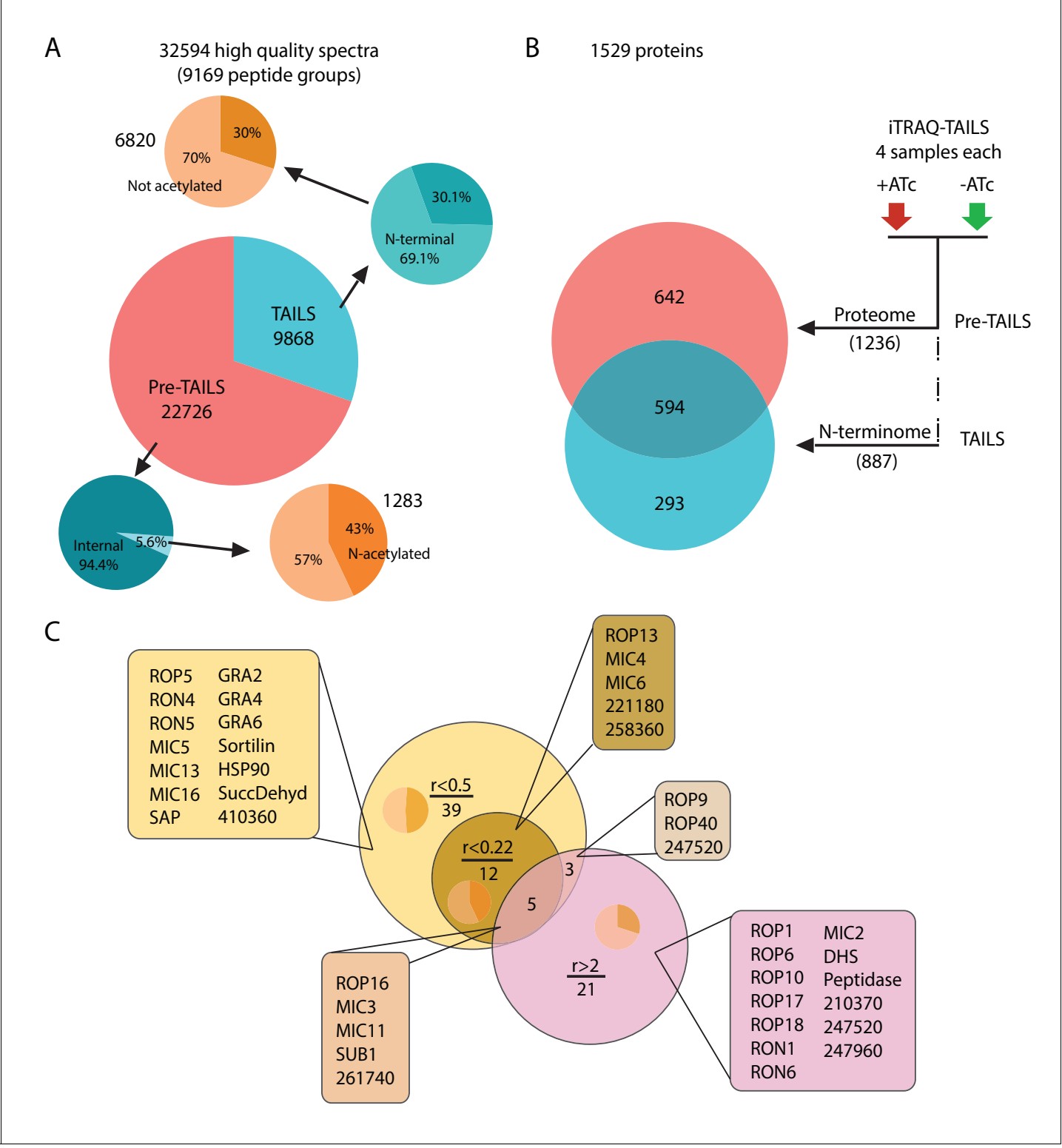

**Figure 2.** Global analysis of MS data and peptide ratios. (**A**) All measured spectra (65'900) minus those with low quality attributes leave 32'594 high quality spectra (9169 peptide groups) for the analysis. The graphical representation depicts proportions of N-terminal peptides in the pre-TAILS and TAILS datasets (blue pie charts) and corresponding N-acetyl modifications (ochre pie charts). N-termini in the TAILS fraction are enriched by >10 fold (69% vs 6%) over internal peptides (generated by trypsin cleavage). (**B**) Graphical representation of the 1529 proteins in the combined datasets. Of those 642 and 293 are unique to the pre-TAILS or the TAILS fractions, respectively. (**C**) Graphical depiction of data generated by calculating +ATc/-ATc peptide ratios revealing ASP3-dependent processing (for details see also Table supplement 1-4). Threshold ratios (r) are indicated in the Venn diagram.
*Figure 2 continued on next page*

*Figure 2 continued*

Lists of proven or predicted secreted proteins represented by peptides with ratios indicated in the circles corresponding to each area in the diagram are boxed. Inset pie charts indicate the proportions of secreted (light sector) vs. cytosolic proteins (dark sector). Gene models for hypothetical proteins are indicated by the ToxoDB 6 digit designator (prefix TGGT1_).

DOI: https://doi.org/10.7554/eLife.27480.013

The following figure supplement is available for figure 2:

**Figure supplement 1.** Schematic of the TAILS assay.

DOI: https://doi.org/10.7554/eLife.27480.014

were still associated (*Figure 4G*), despite the accumulation of unprocessed forms as reported (*Figure 4C–D*).

Like micronemes, the rhoptries contain several proteases that might participate in protein maturation events. Toxolysin-1 (TLN1) is a metalloprotease, localized both to the bulb and the neck of the rhoptries, which is known to undergo a C-terminal cleavage in addition to the removal of its pro-domain mediated by two distinct, uncharacterized maturases (*Figure 4H*) (*Hajagos et al., 2012*). Consistent with this previous observation, ASP3 depletion prevents the pro-domain processing whereas the C-terminal cleavage remains unaffected (*Figure 4I*). Subtilisin 2 (SUB2) is a rhoptry protease previously suspected to participate in the removal of the pro-domains of the rhoptry proteins at a consensus SφXE site (*Figure 4—figure supplements 2* and *3*) (*Miller et al., 2003*). As observed for SUB1, we envisioned that ASP3 could be implicated in the maturation and activation of SUB2 and thus indirectly affected the cleavage of ROPs and RONs. However, endogenous tagging of SUB2 in the background of an inducible ASP3 knockdown strain (SUB2ty/ASP3-iKD) did not reveal any evident defect in the protease maturation (*Figure 4—figure supplement 4A*). To more firmly confirm this result, the disruption of *SUB2* gene was performed with the assistance of CRISPR/Cas9 genome editing and revealed that ROPs and RONs are still processed normally in SUB2-KO parasites (*Figure 4—figure supplement 4B–E*). This excludes the presumed role of SUB2 as a maturase for the ROPs, and suggests instead that ASP3 could be directly responsible for the processing of ROPs and RONs.

## Novel rhoptry proteins identified as ASP3 substrates by the TAILS analysis

To further validate the TAILS analysis, we explored 8 hypothetical proteins (named TAILS1-8) from the TAILS dataset (relaxed stringency, with either >2 and <0.5 ratios), selected based on their expression profile, fitness score (*Sidik et al., 2016*) and conservation (*Table 1*, *Supplementary files 1–3* and *Figure 5—figure supplement 1*). Epitope-tagging of the candidates at the endogenous locus in ASP3myc-iKD revealed that 5 out of the 8 gene products localized to the rhoptries, overlapping with ARO, a known marker of the rhoptries (*Figure 5B,D,F,H,J*). This is consistent with the cell cycle dependent expression profile of their transcripts (*Behnke et al., 2010*) (*Figure 5—figure supplement 1*). While TAILS1 (TGME49_202870) has a nuclear localization, TAILS4 (TGME49_239050) appears to be cytosolic (data not shown). Despite having a typical rhoptry expression profile TAILS2 (TGME49_225860) could not be localized. TAILS3 (TGME49_230350), referred to as a hypothetical protein, corresponds to RON11, a polytopic protein containing a calcium binding EF-hand at its C-terminus recently characterized as a dispensable rhoptry neck protein (*Wang et al., 2016*). Western blot analysis of these tagged TAILS proteins in ASP3myc-iKD +/−ATc revealed defects in processing of all the rhoptry localized proteins and hence confirming them as either direct or indirect ASP3 substrates (*Figure 5C,E,G,I,K*). Intriguingly, in presence of ATc, the unprocessed forms of TAILS3 and TAILS8 (TGME49_321650) exhibit an altered distribution within the organelle (*Figure 5L, M*), but not the TAILS 5, TAILS6 and TAILS7 (*Figure 5—figure supplement 2*). Overall, these results led to the identification of 5 novel rhoptry proteins, validating further the robustness of the TAILS analysis as a powerful technique for protease substrates identification.

**Table 1.** List of peptides detected in the combined TAILS datasets with +ATc/-ATc peptide ratios < 0.22 (green columns) and >2 (red columns).

| ToxoDB gene name | Position in master protein | Annotated peptide sequence | Atc+/Atc- Ratio ≤ 0.22 | (log2) Atc+/Atc- Ratio ≤ −2.18 |
|---|---|---|---|---|
| SUB1 | TGGT1_204050 [45-59] | [E].YQNPTSTYNLIKEIR.[K] | 0.039 | −4.69 |
| SUB1 | TGGT1_204050 [43-59] | [H].GEYQNPTSTYNLIKEIR.[K] | 0.194 | −2.36 |
| PLP1 | TGGT1_204130 [200-220] | [T].APDDDFDFLFEDDTPKKPKSR.[V] | 0.156 | −2.68 |
| MIC11 | TGGT1_204530 [58-68] | [T].EDDKSAASIVR.[G] | 0.045 | −4.48 |
| MIC4 | TGGT1_208030 [45-77] | [D].ITPAGDDVSANVTSSEPAKLDLSCVHSDNKGSR.[A] | 0.186 | −2.42 |
| MIC4 | TGGT1_208030 [58-77] | [T].SSEPAKLDLSCVHSDNKGSR.[A] | 0.19 | −2.4 |
| Hypothetical | TGGT1_212210 [196-220] | [L].FKTGSSENNEVLPSFQDAEKAAPVR.[R] | 0.168 | −2.58 |
| MIC6 | TGGT1_218520 [95-123] | [S].ETPAACSSNPCGPEAAGTCKETNSGYICR.[C] | 0.089 | −3.5 |
| Hypothetical | TGGT1_221180 [471-501] | [A].ADGDSGAGTGSPGETSSKQDSGGVGTKVDAR.[V] | 0.164 | −2.61 |
| Hypothetical (TAILS5) | TGGT1_258360 [83-99] | [Q].SASEADEEEESGGSSKR.[S] | 0.163 | −2.62 |
| Hypothetical | TGGT1_261740 [52-79] | [R].ASHSSSKGEGGDEEKHKDKSPEEGAGDR.[D] | 0.205 | −2.29 |
| ROP16 | TGGT1_262730 [29-40] | [F].EEAQKASEAAKR.[Q] | 0.159 | −2.65 |
| Hypothetical (TAILS6) | TGGT1_273860 [145-166] | [K].QTTKKDEDEDGSEDSEDDEAER.[A] | 0.049 | −4.34 |
| Hypothetical (TAILS6) | TGGT1_273860 [147-166] | [T].TKKDEDEDGSEDSEDDEAER.[A] | 0.079 | −3.66 |
| Hypothetical (TAILS6) | TGGT1_273860 [143-166] | [S].AKQTTKKDEDEDGSEDSEDDEAER.[A] | 0.097 | −3.37 |
| Hypothetical (TAILS7) | TGGT1_279420 [184-205] | [F].SELKSTKSSTAPSDSVKAAATR.[L] | 0.22 | −2.18 |
| Putative HSP75 | TGGT1_292920 [825-849] | [A].EDDKAQPDSSSAQTDSTAGSEVEPR.[K] | 0.141 | −2.82 |
| ROP13 | TGGT1_312270 [67-90] | [E].GTNETNPPTSRPPGWKYEGSDLHR.[R] | 0.219 | −2.19 |
| Hypothetical | TGGT1_315270 [25-45] | [T].LLPSAPKPVDEAALAAAEKER.[E] | 0.19 | −2.4 |
| MIC3 | TGGT1_319560 [57-86] | [F].AVTETHSSVQSPSKQETQLCAISSEGKPCR.[N] | 0.074 | −3.76 |
| MIC3 | TGGT1_319560 [67-86] | [Q].SPSKQETQLCAISSEGKPCR.[N] | 0.079 | −3.65 |
| MIC3 | TGGT1_319560 [69-86] | [P].SKQETQLCAISSEGKPCR.[N] | 0.192 | −2.38 |
| Hypothetical (TAILS8) | TGGT1_321650 [257-268] | [F].AEHKSGGEKASR.[E] | 0.156 | −2.68 |
| **ToxoDB Gene Name** | **Positions in Master Proteins** | **Annotated Peptide Sequence** | **Atc+/Atc- Ratio ≥ 2** | **(log2) Atc+/Atc- Ratio ≥ 1** |
| MIC2 | TGGT1_201780 [27-36] | [G].GGWSIVDALR.[K] | 3.317 | 1.73 |
| SUB1 | TGGT1_204050 [212-225] | [V].NTSSKGSNDPLLDR.[L] | 3.267 | 1.71 |
| SUB1 | TGGT1_204050 [214-225] | [T].SSKGSNDPLLDR.[L] | 3.837 | 1.94 |
| MIC11 | TGGT1_204050 [40-59] | [L].SHHGEYQNPTSTYNLIKEIR.[K] | 4.195 | 2.07 |
| MIC11 | TGGT1_204530 [23-32] | [G].VSEGVVVPVR.[F] | 2.175 | 1.12 |
| MIC11 | TGGT1_204530 [25-32] | [S].EGVVVPVR.[F] | 3.548 | 1.83 |
| ROP18 | TGGT1_205250 [50-72] | [T].LGPSKLDSKPTSLDSQQHVADKR.[W] | 4.985 | 2.32 |
| Hypothetical | TGGT1_210370 [65-89] | [N].NPPPLEGASVSPENATDPPETGGSR.[R] | 2.824 | 1.5 |
| Methionine aminopeptidase | TGGT1_211330 [338-359] | [C].SSPDVSSDNASSSTDLSFPVLR.[R] | 2.033 | 1.02 |

*Table 1 continued on next page*

*Table 1 continued*

| ToxoDB gene name | Position in master protein | Annotated peptide sequence | Atc+/Atc-Ratio ≤ 0.22 | (log2) Atc+/Atc-Ratio ≤ −2.18 |
|---|---|---|---|---|
| Hypothetical (TAILS3) | TGGT1_230350 [134-141] | [K].LDNPELSR.[Q] | 3.549 | 1.83 |
| IMC1 | TGGT1_231640 [1-20] | [-].MFKDCADPCSDCCQPAEQQR.[G] | 2.028 | 1.02 |
| IMC1 | TGGT1_233820 [1-27] | [-].MKPPSGLSGASAQGVGAEETSVSLLAR.[L] | 2.028 | 1.02 |
| IMC1 | TGGT1_235620 [13-49] | [M].VMPASQGAPHGAIAAESQEKTNSCVSQECPASSETAR.[Q] | 2.466 | 1.3 |
| Peptidase M16 | TGGT1_236210 [36-53] | [G].FFSAAPAAATAGVSPLAR.[S] | 2.005 | 1 |
| Hypothetical (TAILS4) | TGGT1_239050 [432-438] | [P].LSPPDSR.[G] | 5.526 | 2.47 |
| ROP9 | TGGT1_243730 [59-71] | [P].QGSPPASQKEAIR.[D] | 2.275 | 1.19 |
| Hypothetical | TGGT1_247520 [44-58] | [Q].NPAGGKGGSGPHGGR.[R] | 2.431 | 1.28 |
| Hypothetical | TGGT1_247520 [36-58] | [A].SDQKQGSQNPAGGKGGSGPHGGR.[R] | 6.971 | 2.8 |
| Hypothetical | TGGT1_247960 [43-52] | [L].LPGDPVLFPR.[S] | 9.446 | 3.24 |
| ROP17 | TGGT1_258580 [26-42] | [R].SPTSNDVFGELVASAER.[A] | 12.98 | 3.7 |
| ROP6 | TGGT1_258660 [65-79] | [K].GSDFGEVKLGSAGQR.[Q] | 6.897 | 2.79 |
| Hypothetical | TGGT1_261740 [56-79] | [S].SSKGEGGDEEKHKDKSPEEGAGDR.[D] | 3.973 | 1.99 |
| Hypothetical | TGGT1_261740 [54-79] | [S].HSSSKGEGGDEEKHKDKSPEEGAGDR.[D] | 5.053 | 2.34 |
| Hypothetical | TGGT1_261740 [56-79] | [S].SSKGEGGDEEKHKDKSPEEGAGDR.[D] | 11.39 | 3.51 |
| ROP16 | TGGT1_262730 [27-40] | [M].SFEEAQKASEAAKR.[Q] | 6.953 | 2.8 |
| Hypothetical | TGGT1_268835 [152-160] | [R].DSWMSLAPF.[V] | 2.14 | 1.1 |
| SRS30C | TGGT1_273120 [53-79] | [A].AKAQGGETPPSDPTCVVEGAVTKCTCR.[N] | 3.009 | 1.59 |
| Hypothetical | TGGT1_276190 [1-31] | [-].MSPADPEAGSLQSSAPPLASAGKSAGAGAPR.[A] | 2.081 | 1.06 |
| ROP40 | TGGT1_291960 [53-67] | [A].TDSDSEPEGKGGYQR.[L] | 6.511 | 2.7 |
| ROP40 | TGGT1_291960 [55-67] | [D].SDSEPEGKGGYQR.[L] | 7.127 | 2.83 |
| Hypothetical | TGGT1_295360 [1-29] | [-].MEQQQDELKHSWGANELPAGQQGSPLAER.[Q] | 2.309 | 1.21 |
| RON6 | TGGT1_297960A [50-56] | [L].LPAGQDR.[S] | 3.172 | 1.67 |
| ROP1 | TGGT1_309590 [274-296] | [L].LEPTEEQQEGPQEPLPPPPPPTR.[G] | 2.111 | 1.08 |
| ROP1 | TGGT1_309590 [58-76] | [G].VPAYPSYAQVSLSSNGEPR.[H] | 4.051 | 2.02 |
| ROP1 | TGGT1_309590 [87-115] | [M].SVKPHANADDFASDDNYEPLPSFVEAPVR.[G] | 4.094 | 2.03 |
| ROP1 | TGGT1_309590 [103-115] | [N].YEPLPSFVEAPVR.[G] | 10.23 | 3.36 |
| RON1 | TGGT1_310010 [66-82] | [R].AAANGSEGGVAQSEQER.[A] | 3.193 | 1.67 |
| ROP10 | TGGT1_315490 [35-48] | [R].ESPQWDSLLPLQDR.[R] | 16.32 | 4.03 |
| MIC3 | TGGT1_319560 [60-86] | [T].ETHSSVQSPSKQETQLCAISSEGKPCR.[N] | 4.882 | 2.29 |
| MIC3 | TGGT1_319560 [56-86] | [S].FAVTETHSSVQSPSKQETQLCAISSEGKPCR.[N] | 9.061 | 3.18 |
| MIC3 | TGGT1_319560 [64-86] | [S].SVQSPSKQETQLCAISSEGKPCR.[N] | 11.35 | 3.5 |
| MIC3 | TGGT1_319560 [52-86] | [S].LAPSFAVTETHSSVQSPSKQETQLCAISSEGKPCR.[N] | 34.74 | 5.12 |

DOI: https://doi.org/10.7554/eLife.27480.015

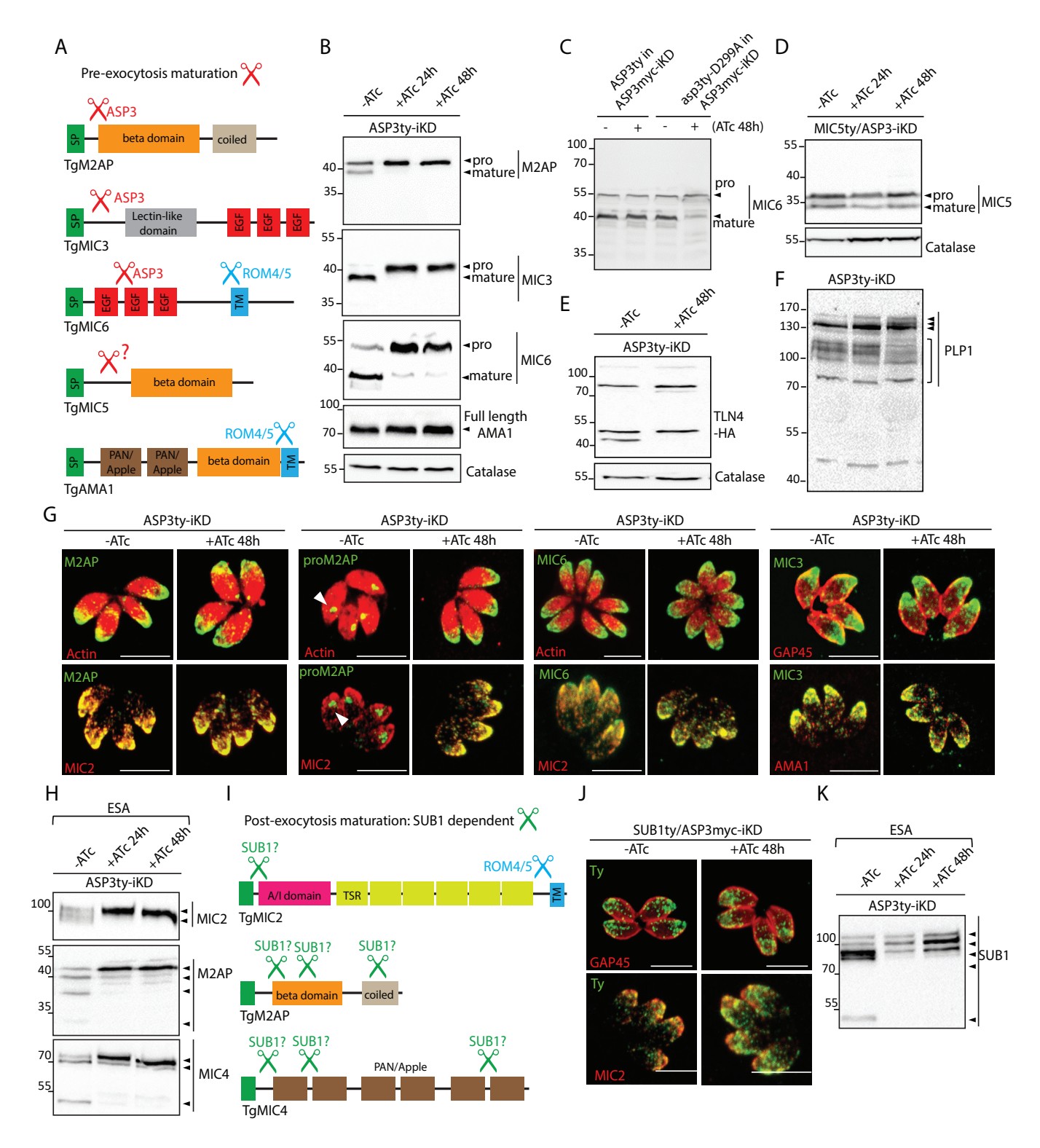

**Figure 3.** ASP3 is a maturase for microneme proteins and impacts microneme processing post-exocytosis via SUB1. (**A**) Schematic representation of MICs and their processing pre-exocytosis (red) and post-exocytosis by ROM4/5 (light blue). (**B**) Immunoblots assessing the processing of the microneme proteins M2AP, MIC3, MIC6, AMA1 upon ASP3 knockdown. Catalase was used as loading control. (**C**) ASP3ty but not asp3ty-D299A can rescue the processing of MIC6 in absence of ASP3myc. (**D**), (**E**), (**F**) Immunoblots assessing the processing of the microneme proteins MIC5, TLN4 and PLP1 upon ASP3 knockdown. MIC5 was Ty-tagged at the endogenous locus in ASP3-iKD and transient plasmid transfection of epitope tagged TLN4 was used. (**G**)

*Figure 3 continued on next page*

*Figure 3 continued*

IFAs evaluating the localization of microneme proteins upon ASP3 depletion – M2AP, MIC6 and MIC3, co-localized with either MIC2 or AMA1. White arrowhead points to the pro form of M2AP. (H) Immunoblot analyses of ESA from ASP3ty-iKD parasites, +/−ATc, after 2% EtOH stimulation show impaired post-exocytosis cleavage of MIC2, M2AP and MIC4. (I) Schematic representation of microneme proteins and their putative processing post-exocytosis by SUB1. (J) Endogenously tagged SUB1 was localized in the microneme independently of ASP3. (K) Depletion of ASP3 showed impaired processing of SUB1in ESA analyses. TgSUB1 was detected using an antibody raised against PfSUB1. Black arrowheads represent pro and mature forms of the proteins.

DOI: https://doi.org/10.7554/eLife.27480.016

The following figure supplement is available for figure 3:

**Figure supplement 1.** Effect of ASP3 depletion on MICs.

DOI: https://doi.org/10.7554/eLife.27480.017

## The ethylamine scaffold based compound 49c recapitulates ASP3 depletion phenotypes

Aspartyl proteases qualify as attractive targets for chemotherapy against pathogens and have been the focus of intense research toward the identification of antimalarial drugs. Interestingly, 49c belongs to a series of antimalarials based on a hydroxyethylamine scaffold, and designed to inhibit the malaria Plasmepsin II (*Boss et al., 2003*; *Ciana et al., 2013*) (*Figure 6A*). 49c exhibits a subnano-molar IC50 against *P. falciparum* and acts on the late erythrocytic stages by blocking egress and invasion, potentially targeting the two late schizont aspartyl proteases, PMIX and PMX, that cluster phylogenetically with ASP3 (*Figure 1—figure supplement 1*). Given the significant sequence conservation between the members of this cluster (*Figure 1—figure supplement 2B*), we hypothesized that 49c could target ASP3 and impact on *T. gondii* invasion and egress. Concordantly, 49c interferes with the lytic cycle of *T. gondii* tachyzoites, with an IC50 of ~676 nM (*Figure 6B*). A deeper analysis confirmed that 49c blocks invasion, egress and rhoptry discharge without impacting on gliding motility, intracellular growth and microneme secretion (*Figure 6C–G* and *Figure 6—figure supplement 1A–B*). Moreover, 49c accurately recapitulates ASP3 depletion by inhibiting MICs, ROPs and RONs pre-exocytosis as well as some post-exocytosis processing events of MICs, without affecting their trafficking to the respective secretory organelles (*Figure 6E,F,H* and *Figure 6—figure supplement 1C–D*). In contrast, 49b, another hydroxyethylamine scaffold derivative having poor efficacy against Plasmodium (*Ciana et al., 2013*) showed concordant low inhibitory activity against tachyzoites (IC50 >10 µM) (*Figure 6A,B,H*). We took advantage of the inhibitory effect of 49c on ASP3 to narrow down the time of action of ASP3 during biogenesis and maturation of the secretory organelles. A minimal impact on invasion and egress was measured when the parasites were treated with 49c either in extracellular condition or less than 3 hr prior to natural egress or calcium ionophore stimulated egress (*Figure 6I–K*). Comparatively, 49c mediated defect in invasion or induced egress was highest when the parasites were pre-treated for 42 to 48 hr (*Figure 6I–K*). Moreover, washing the compound just 6 hr or even only 3 hr prior to inducing the egress failed to rescue the phenotype, resulting in no significant increase in the number of lysed vacuoles due to egress (*Figure 6J–K*). This is consistent with a block in the maturation of the MICs, ROPs and RONs that are produced and processed during the 2–3 hr period dedicated to the biogenesis of the secretory organelles (*Nishi et al., 2008*). Taken together, the time frame of parasite sensitivity to 49c reflects the time necessary for the maturation and trafficking of the MICs/RONs/ROPs prior to reaching their respective organelles.

In *T. gondii* tachyzoites, the other expressed ASP1, ASP2 and ASP5, are dispensable. Importantly, ASP5 plays a key role in the formation of the intravacuolar membranous nanotubular network (MNN) (*Hammoudi et al., 2015*; *Coffey et al., 2015*). However, ASP3 depleted parasites and 49c treated parasites are not affected in MNN formation (*Figure 6—figure supplement 2A*). Moreover 49c has no impact on the maturation of ASP5 or on the processing and export of GRA16, an ASP5 substrate, inferring that 49c does not inhibit ASP5 (*Figure 6—figure supplement 2B–D*).

## In vitro cleavage activity of ASP3 and inhibition by 49c

To formally demonstrate that ASP3 acts as secretory protein maturase, we immunoprecipitated ASP3ty from extracellular parasites under native conditions and assessed its proteolytic activity

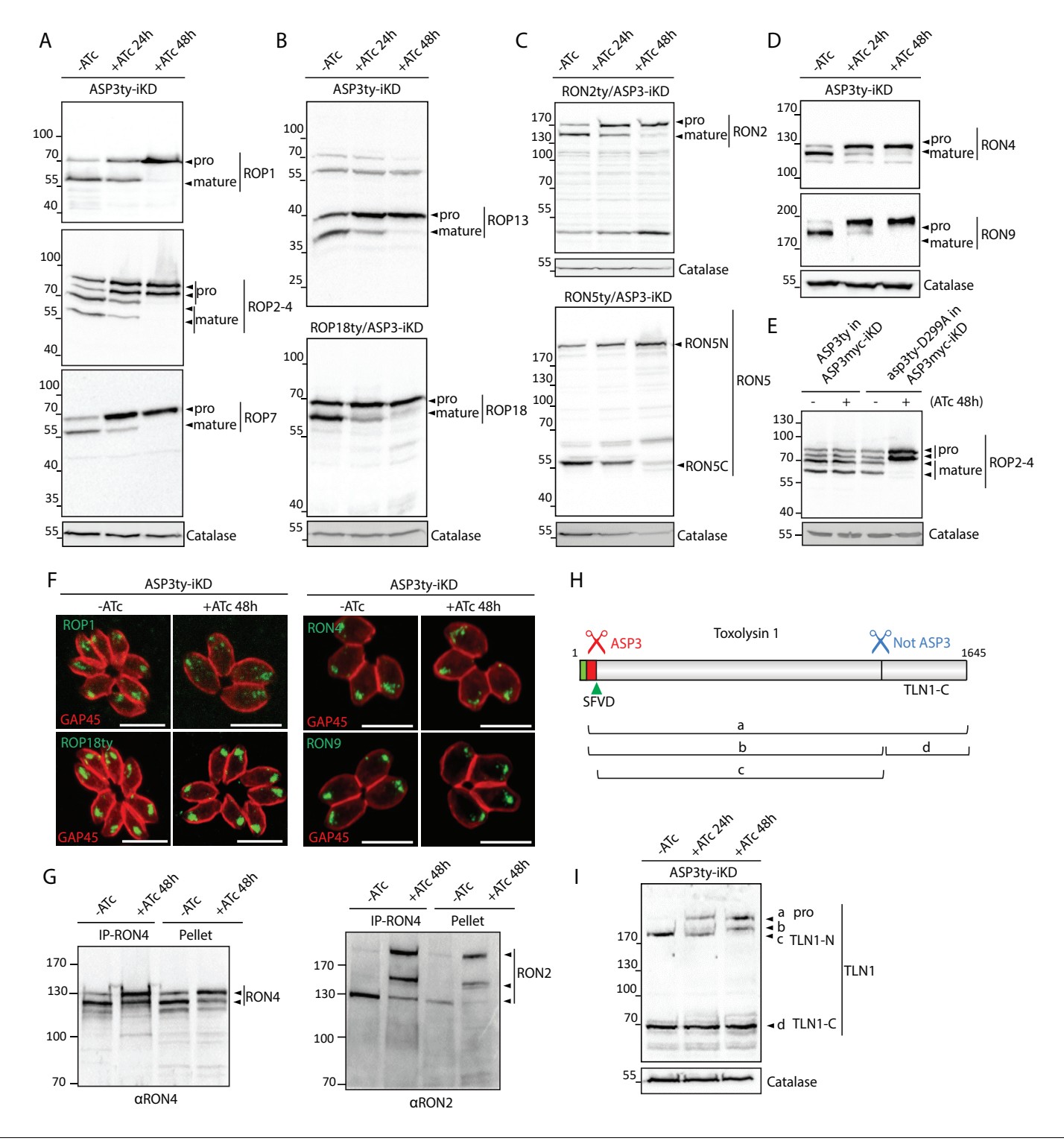

**Figure 4.** ASP3 is involved in the processing of rhoptry proteins. (**A**), (**B**) Immunoblots evaluating the processing of the rhoptry bulb proteins ROP1, ROP2-4, ROP7, ROP13, and ROP18 upon ASP3 knockdown. ROP18 was Ty-tagged at endogenous locus in ASP3-iKD. Catalase was used as a loading control. Arrowheads represent pro and mature forms of the proteins. (**C**), (**D**) Immunoblots evaluating the processing of the rhoptry neck proteins RON2, RON4, RON5 and RON9 upon ASP3 knockdown. RON2 and RON5 were Ty-tagged at the endogenous locus in ASP3-iKD. (**E**) ASP3ty but not asp3ty-D299A can rescue the processing of ROP2-4 in absence of ASP3myc. (**F**) IFAs evaluating localization of ROP1, ROP18, RON4, and RON9 upon ASP3 knockdown. No alteration was observed in absence of ASP3. (**G**) ASP3 depletion did not impact on the formation of the RON4-RON2 complex as

*Figure 4 continued on next page*

*Figure 4 continued*

demonstrated by co-immunoprecipitation of RON2 with RON4 antibodies. (**H**) Schematic for the processing events and products of TLN1. (**I**) Immunoblot evaluating TLN1 showed that ASP3 depletion abolished the pro-domain processing of TLN1.

DOI: https://doi.org/10.7554/eLife.27480.018

The following figure supplements are available for figure 4:

**Figure supplement 1.** Effect of ASP3 depletion on rhoptry proteins.

DOI: https://doi.org/10.7554/eLife.27480.019

**Figure supplement 2.** SφXE motifs in ROPs.

DOI: https://doi.org/10.7554/eLife.27480.020

**Figure supplement 3.** SφXE motifs in RONs.

DOI: https://doi.org/10.7554/eLife.27480.021

**Figure supplement 4.** SUB2 has no impact on the processing of ROPs/RONs/MICs.

DOI: https://doi.org/10.7554/eLife.27480.022

(*Figure 7—figure supplement 1A*). The cleavage site of MIC6 was previously mapped after Ser94 (VQLS|ETPA) (*Meissner et al., 2002*), and this position was also confirmed by the TAILS analysis (*Table 1* and *Supplementary files 1* and *2*). Firstly, a bacterial recombinant fragment of MIC6 encompassing the first three EGF domains (GST-MIC6) including the VQLS|ETPA cleavage site was used as substrate. GST-MIC6 was cleaved efficiently by pulled-down ASP3ty while no processing was detectable using the catalytically dead asp3ty-D299A (*Figure 7A*). These results were confirmed with a fluorogenic peptide, 10 amino acids long, comprising the predicted cleavage site (*Figure 7C–D*). Again no processing was detectable using asp3ty-D299A, confirming that the observed activity is specific to ASP3ty (*Figure 7D*).

To confirm that 49c acts as selective inhibitor for ASP3, this compound as well as 49b were tested on the MIC6 peptide and GST-MIC6 in *in vitro* cleavage assays. 49c blocked the cleavage of GST-MIC6 cleavage at 1 µM and of MIC6 peptide cleavage at a concentration of 100 nM, whereas pepstatin A, a potent and reversible inhibitor of aspartic proteases of microbial origin, had no inhibitory effect on ASP3 activity even at 10 µM (*Figure 7B,D*). To further support the broadness of ASP3 substrate specificity, additional fluorogenic peptides were tested (*Figure 7E–F*). ROP1 peptide was chosen based on its known cleavage site (*Bradley and Boothroyd, 1999*; *Soldati et al., 1998*) whereas MIC3 and ROP13 peptides were based on the deduced cleavage site from the TAILS analysis. SUB1 peptide was chosen based on the putative SφXE cleavage site (*Figure 7E–F*). In the absence of experimental data, a RON4 peptide was designed based on *in silico* prediction of a putative cleavage site and deduced cleavage site from TAILS data (*Figure 7—figure supplement 1B* and *Supplementary file 2*). The results confirmed that ASP3 acts as a maturase for MICs and ROPs whereas the data on RONs remain inconclusive given the uncertainty about the cleavage site. 49c consistently blocks ASP3 activity on MIC and ROP substrates, compellingly supporting that its inhibitory effect on the parasite lytic cycle is mediated through ASP3 inhibition.

## Discussion

The large repertoire of regulated secretory proteins stored in micronemes critically participate in invasion and egress while rhoptry proteins are dedicated to invasion, PVM formation and subversion of host cellular functions. MICs are processed prior to exocytosis by uncharacterized proteases and post-exocytosis by subtilisin and rhomboid-like proteases. Even less in known about the processing of RONs and ROPs and the protease(s) implicated. Earlier studies have postulated the contribution of CPL and SUB2 towards the processing of MICs and ROPs respectively (*Miller et al., 2003*; *Parussini et al., 2010*), however, in both cases, either limited or no direct evidence were supporting the claims.

We have demonstrated here that ASP3 is a resident aspartyl protease of the ELC, critical for tachyzoites invasion and egress but not implicated in parasite replication, gliding motility and host cell attachment. To explain ASP3 depletion phenotypes at the molecular level, we applied for the first time the power of TAILS analysis to generate in-depth characterization of ASP3 dependent protein N-termini in *T. gondii*. TAILS pipeline is based on negative selection, where native as well as proteolytically generated protein N-termini are identified and quantified. A comparative analysis

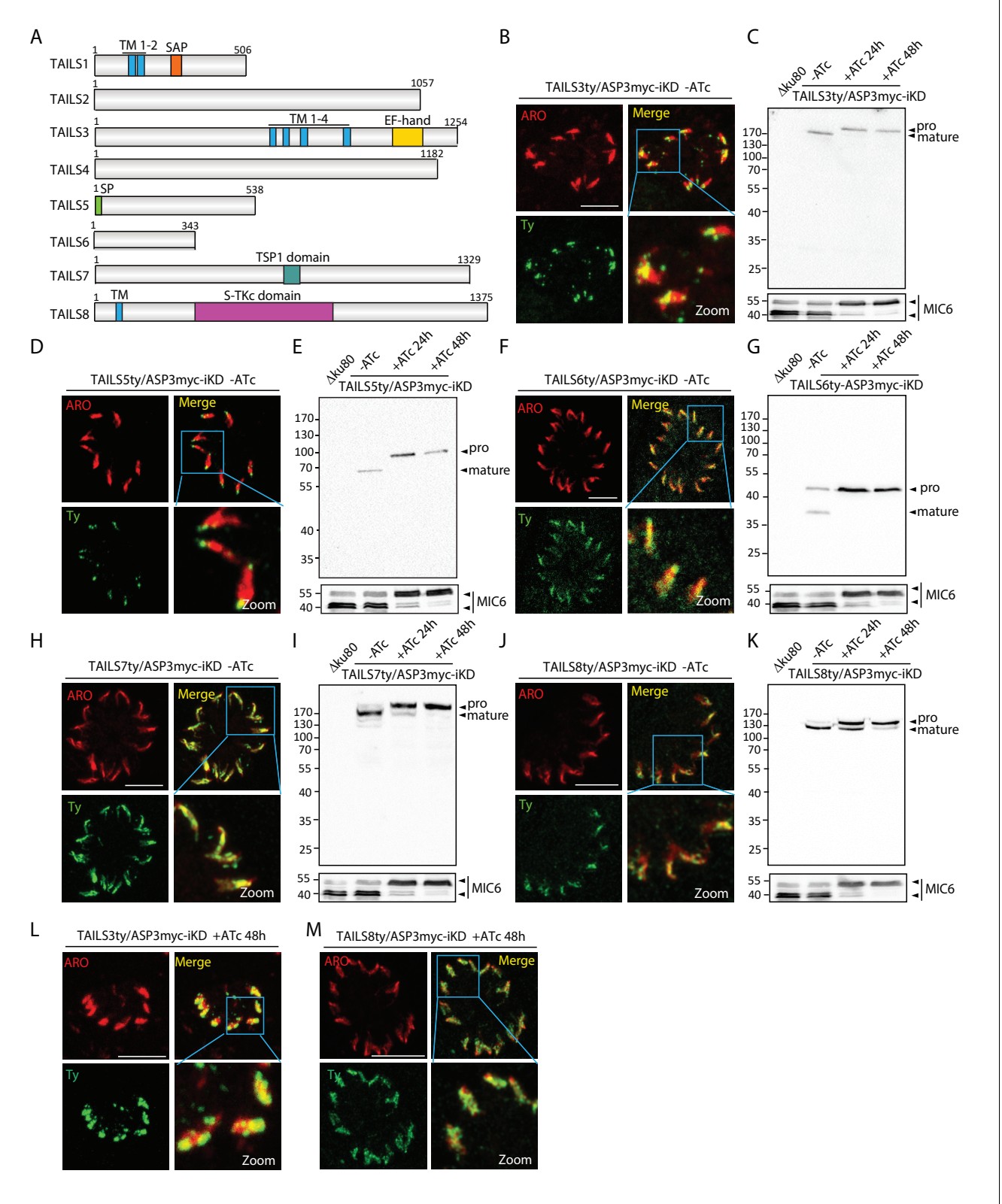

**Figure 5.** TAILS analysis identified new rhoptry proteins as ASP3 substrates. (**A**) Schematic representation of the different candidate identified during the TAILS analysis. SP – signal peptide, TM – transmembrane, SAP - SAF-A/B, Acinus and PIAS domain, TSP1 - thrombospondin-1, S-TKc – Serine-Threonine kinase catalytic domain. (**B**), (**D**), (**F**), (**H**), (**J**) TAILS 3, 5, 6 7 and 8 were localized to the rhoptries and (**C**), (**E**), (**G**), (**I**) (**K**) showed impaired

*Figure 5 continued on next page*

Figure 5 continued

processing upon ASP3 depletion. Arrowheads represent pro and mature forms of the proteins. MIC6 was used as control. (**L, M**) Altered localization of TAILS3 and TAILS8 in the rhoptries upon ASP3 depletion.

DOI: https://doi.org/10.7554/eLife.27480.023

The following figure supplements are available for figure 5:

**Figure supplement 1.** Expression profile of TAILS candidates.

DOI: https://doi.org/10.7554/eLife.27480.024

**Figure supplement 2.** Effect of ASP3 knockdown on the localization of the TAILS candidates.

DOI: https://doi.org/10.7554/eLife.27480.025

between WT and ASP3-depleted parasites identified protease cleavage sites dependent on ASP3. The obtained TAILS data document many proteolytic processing events accurately but as a method also highlight secondary effects of ASP3 knockdown with considerable sensitivity. The most frequent case is detection of peptides representing degradation products that increase in +ATc conditions presumably due to accumulation of unprocessed precursors, and in some cases temporally and spatially incorrect trafficking (*Prudova et al., 2016*).

The criteria applied for the identification of ASP3-dependent processing events are +ATc/-ATc ratios of <0.22 (0.5) for N-terminal peptides in conjunction with the detection of additional peptides representing immature or degraded products with ratios >2. The over-representation of MICs, ROPs and RONs as ASP3 candidate substrates was compelling. Taken together, the results of the TAILS analysis to detect ASP3-dependent substrates show that interpretation of peptide ratio profiles can be relatively straightforward in some cases (e.g. MIC3, 11), but requires more in-depth analysis of the MS data in others (e.g. MIC6, TGGT1_273860 – TAILS3) (Appendix 1). Even though TAILS can never be comprehensive in terms of defining protease-specific substrates in knockdown parasites, a detailed analysis of the data was instrumental to inform about new targets for experimental testing.

The TAILS analysis confirms the previously reported cleavage site of MIC6 (VQLS|ETPA), ROP1 (SFVE|APVR) and ROP13 (SFTE|GTNE). Similarly, several peptides corresponding to cleavage sites around the putative rhoptry cleavages site, SφXE, were detected, confirming the reliability of the experimental procedure and highlighting the strength of TAILS N-terminomics. The purified ASP3 was used to firmly prove that MIC6, MIC3, ROP1 and ROP13 are ASP3 substrates based on enzymatic cleavage assay on the corresponding fluorogenic peptides (*Figure 7D–F*). However, the data did not reveal an obvious consensus sequence motif for ASP3-specific cleavage sites. It is likely that substrate recognition and cleavage are governed by both secondary structure features and interactions between the amino acids surrounding the substrate cleavage site and the protease. It should also be noted that the many cleavage events and enriched peptides could possibly be due to secondary effects caused by altered abundance or activation of other proteases upon ASP3 depletion. The combination of *in vivo* and *in vitro* data demonstrates the role of ASP3 in the pre-exocytosis processing of apical organellar proteins – MICs, ROPs. While no direct evidence was obtained to demonstrate that ASP3 processes the RONs, it is likely to be the case.

In parallel, we also obtained experimental evidence for ASP3-dependent processing in the absence of TAILS supporting data. Importantly, it was possible to show that the pattern of SUB1 processing is altered in absence of ASP3, which hampers its activation and thus explains the indirect impact of ASP3 depletion on the post-exocytosis processing of MIC2, M2AP and MIC4 (*Figure 3G– J*).

Depletion of ASP3 causes a severe block both in invasion and egress that could be attributed to a defect in rhoptry discharge and in host plasma membrane lysis, respectively. As reported earlier, the cleavage per se is not essential for trafficking of organellar proteins to their respective organelles. In consequence, the phenotypic links between an absence of proteolytic processing of ROP/ RON/MIC proteins and the observed phenotypes are not obvious. Both microneme exocytosis and rhoptry discharge occur at the apex of the parasite during invasion. While microneme secretion is driven by changes in cytoplasmic calcium concentration and by phosphatidic acid production (*Bullen et al., 2016*) that can be triggered by various chemicals such as ethanol, the trigger for rhoptry secretion is still elusive (*Lourido and Moreno, 2015*; *Sharma and Chitnis, 2013*). At the time of invasion, the parasite's apical tip comes in close contact with the host cell membrane leading to rhoptry discharge. The secreted rhoptry material fuses with the invaginating host cell membrane and

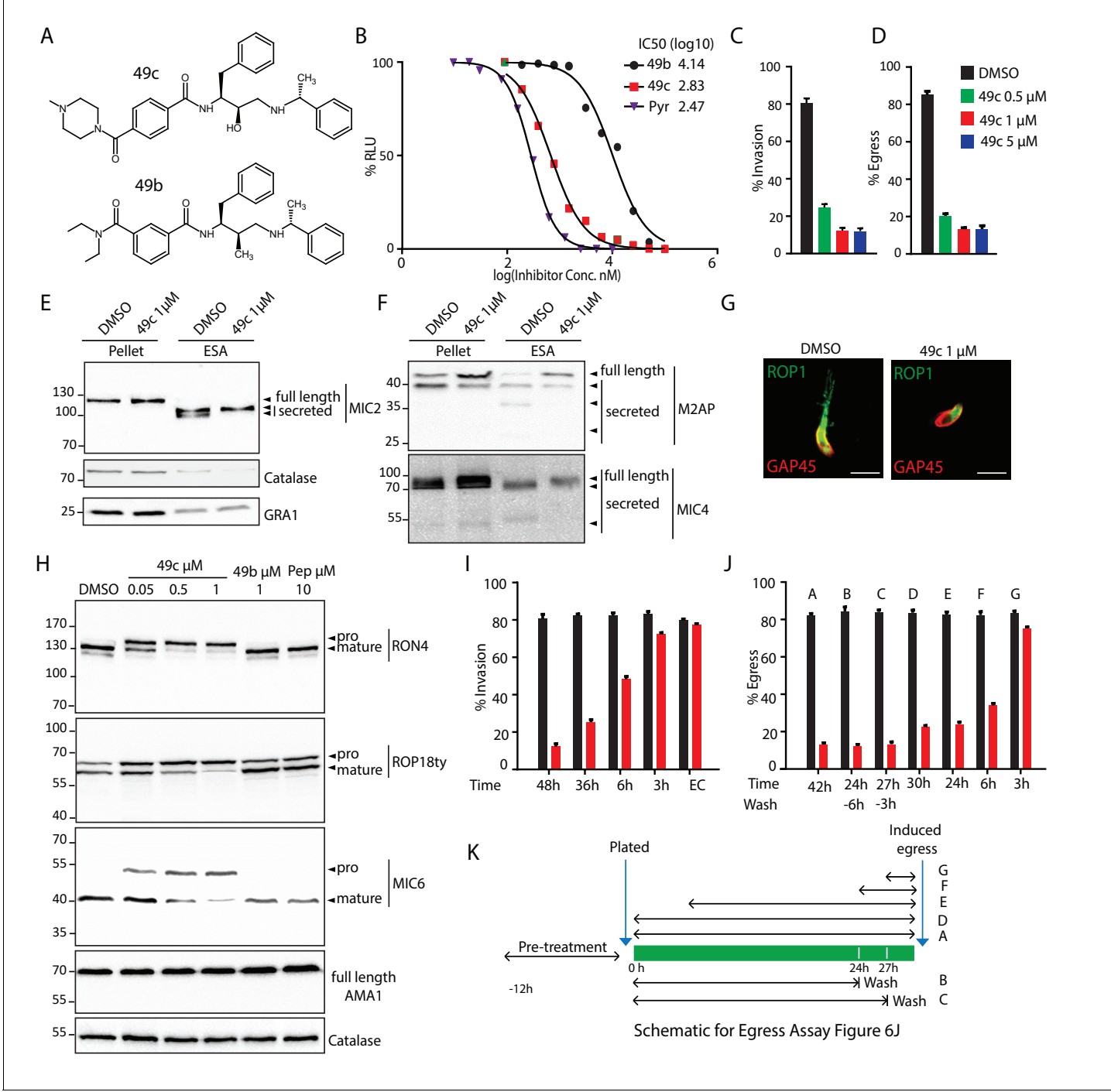

**Figure 6.** 49c recapitulates the phenotype of ASP3 depletion in *T. gondii*. (**A**) Chemical structure of compounds 49b and 49c. (**B**) IC50 on *T. gondii* RH-pTub-CBG99-luciferase parasites of 49b (13.8 µM), 49c (676 nM) and Pyrimethamine (295 nM), used here as a positive control. (**C**) 49c treatment impaired RH parasites invasion and (**D**) egress. Data are presented as mean ±standard deviation (SD) from 3 independent experiments. (**E, F**) Parasites, treated for 36 hr with DMSO or 49c, showed normal secretion of microneme proteins (MIC2, M2AP, MIC4), but altered processing in this fraction (MIC2, M2AP, MIC4). Arrowheads show full length and secreted fragments. Catalase was used as cytosolic control and GRA1 for constitutive secretion. (**G**) 49c treated during 48 hr of Δku80 parasites displayed a complete absence of rhoptry content secretion as assessed by probing against the rhoptry protein ROP1. (**H**) Processing of RON4, ROP18 and MIC6 was affected by 48 hr of 49c treatment while DMSO treated samples were not affected. AMA1 serves a control for an unprocessed microneme and Catalase as loading control. ROP18 was Ty-tagged at endogenous locus. Pepstatin was used as a negative control. (**I**) Invasion was blocked by 49c when parasites were treated for 6 hr or more prior to egress, but not less than 3 hr or when

*Figure 6 continued on next page*

*Figure 6 continued*

extracellular. (J) Parasites were blocked in egress when treated with 49c at least 3 hr before induced egress, but not later. (K) Schematic showing the various treatments for the induced egress assay in J.

DOI: https://doi.org/10.7554/eLife.27480.026

The following source data and figure supplements are available for figure 6:

**Source data 1.** Source data of the triplicate experiments done on RHΔku80 parasites (DMSO or 49c treated) Intracellular growth assay (*Figure 6—figure supplement 1A*), Invasion assay (C), Egress assay (D), Invasion assay with washes (I) and Egress assay with washed (J).
DOI: https://doi.org/10.7554/eLife.27480.029
**Source data 2.** Source data of the in vitro measurement of IC50 done on luciferase expressing *T. gondii* tachyzoites (*Figure 6B*).
DOI: https://doi.org/10.7554/eLife.27480.030
**Figure supplement 1.** 49c has no impact on parasite intracellular replication, gliding motility, and on the localization of rhoptry and microneme proteins.
DOI: https://doi.org/10.7554/eLife.27480.027
**Figure supplement 2.** 49c has no impact on ASP5 maturation, intravacuolar membranous nanotubular network (MNN) formation or on the maturation and export of ASP5 substrate, GRA16.
DOI: https://doi.org/10.7554/eLife.27480.028

forms the PVM. To date, only two proteins have been associated to rhoptry discharge – ARO, which is implicated in the apical positioning of the rhoptries (*Mueller et al., 2013*) whereas the role of type I transmembrane microneme protein MIC8 is not understood (*Kessler et al., 2008*). Here we show that the rhoptries are apically positioned and MIC8 is not affected by ASP3 depletion, which leaves open the identification of the ASP3 substrate implicated in rhoptry discharge. In contrast, PLP1 is an obvious candidate to explain the defect in egress observed in ASP3 depleted parasites. PLP1 was identified in the TAILS analysis and WB performed in absence of ASP3 indicates slight alternation of the processing pattern that might result in a partial dysfunction of the perforin activity (*Roiko and Carruthers, 2013*).

The TAILS dataset identified several hypothetical proteins and 5 out of the 8 candidates analyzed further, were confirmed as bona fide rhoptry proteins and ASP3 substrates. Functional characterization of these and other TAILS candidates should help identifying the factor critical for the initial step of invasion and implicated in rhoptry discharge.

Invasion and egress are key events for survival and dissemination of the Apicomplexa and an aspartic protease essential for these two steps constitutes an attractive target for drug intervention. Antimalarials primarily target apicoplast or food vacuole that are hosting metabolic pathways and ensuring parasite growth. The Plasmodium aspartic proteases in particular have been explored for specific inhibitors, with a main focus on the food vacuole Plasmepsins implicated in haemoglobin degradation (*Figure 1—figure supplement 1*). However, a considerable redundancy among the proteases involved render them sub-optimal targets. We report here on an aspartyl protease inhibitor that acts at subnanomolar IC50 on *P. falciparum* and presumably targets the Plasmepsins IX and X that belong to the same phylogenetic cluster as ASP3 (*Figure 1—figure supplement 1*). 49c is a hydroxyethylamine scaffold based compound that kills *T. gondii* with an IC50 of 676 nM by blocking invasion (rhoptry discharge) and egress, without impacting on gliding motility, intracellular growth or microneme secretion. 49c recapitulates the ASP3 depletion phenotypes and inhibits the processing of ASP3 substrates. 49c also compellingly blocked ASP3 mediated cleavage of substrates (MIC6, ROP1, ROP13, SUB1, MIC3) *in vitro*, thus confirming that it targets ASP3 with high specificity.

In conclusion, we characterized a *T. gondii* aspartyl protease, ASP3, essential for parasite egress and discharge of rhoptry content, thereby, invasion. TAILS analysis and subsequent validation led to define the repertoire of ASP3 substrates. The peptidomimetic inhibitor 49c validates the druggability of ASP3 to interfere with two key steps in the lytic cycle of this apicomplexan parasite.

## Materials and methods

### *Toxoplasma*, host cell and bacteria culture

*T. gondii* tachyzoites, parental and modified strains, were grown in confluent monolayers of human foreskin fibroblasts (HFFs) in Dulbecco's Modified Eagle's Medium (DMEM, Gibco) supplemented

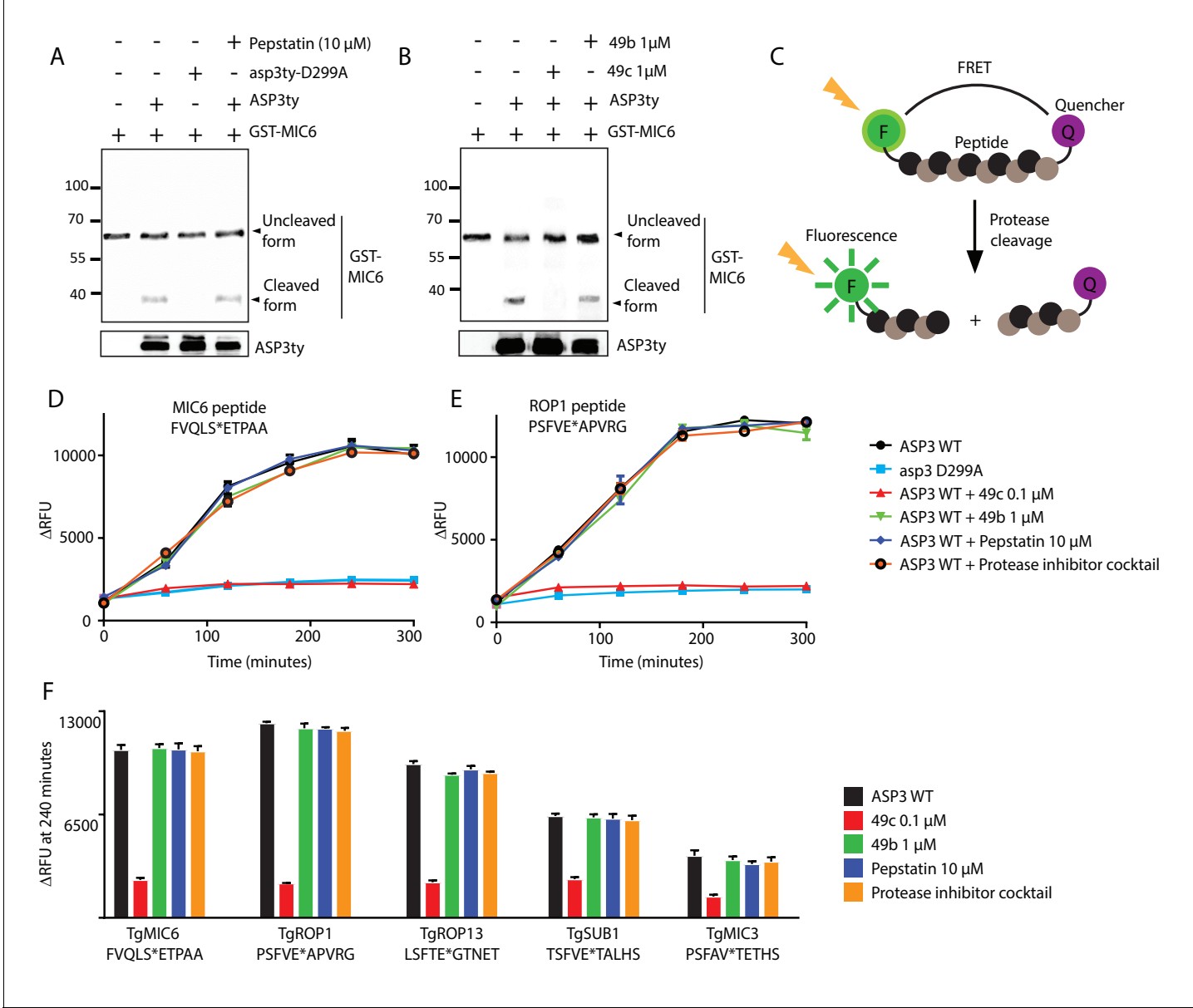

**Figure 7.** In vitro cleavage of ASP3 substrates. (**A**) Recombinant GST-MIC6 was processed by ASP3ty but not by asp3ty-D299A. Pepstatin A was used as negative control. (**B**) 49c but not 49b prevented GST-MIC6 processing by ASP3ty. ASP3 WT and asp3ty-D299A mutant were immunoprecipitated from the parasite. (**C**) Schematic representation of the fluorogenic peptide cleavage assays. (**D**) Cleavage of MIC6 fluorogenic peptide by ASP3ty and asp3ty-D299A in the presence or absence of 49b, 49c, pepstatin and protease cocktail inhibitor. (**E**) Cleavage of ROP1 fluorogenic peptide by ASP3ty and asp3ty-D299A in the presence or absence of 49b, 49c, pepstatin and protease cocktail inhibitor. (**F**) Quantification of peptide cleavage assays for MIC6, ROP1, ROP13, SUB1 and MIC3 at 240 min in the presence or absence of 49b, 49c, pepstatin or protease cocktail inhibitor.

DOI: https://doi.org/10.7554/eLife.27480.031

The following source data and figure supplement are available for figure 7:

**Source data 1.** Source data for the in vitro cleavage assays on fluorogenic peptides for MIC6 (D), ROP1 (E), MIC6/ROP1/ROP13/SUB1/MIC3 at 240 min (F) and RON4/MIC3 (*Figure 7—figure supplement 1B*).

DOI: https://doi.org/10.7554/eLife.27480.033

**Figure supplement 1.** In vitro cleavage assays on ASP3 putative substrates.

DOI: https://doi.org/10.7554/eLife.27480.032

with 5% fetal calf serum (FCS), 2 mM glutamine and 25 mg/ml gentamicin. *E.coli* XL-10 Gold chemically competent bacteria were used for all recombinant DNA experiments.

## Antibodies

The following antibodies were used in this study: T34A11 mAb anti-MIC2 (*Achbarou et al., 1991*), rabbit anti-AMA1 (*Lamarque et al., 2014*), rabbit anti-GAP45, mAb anti-Myc (9E10), mAb anti-Ty1 (BB2), mAb anti-GRA1 (*Sibley et al., 1995*), rabbit anti-MIC6 (*Reiss et al., 2001*), rabbit anti-MIC4 (*Reiss et al., 2001*), mAb T52A3 anti-ROP1 (unpublished), ARO (*Mueller et al., 2013*), ATrx (*DeRocher et al., 2008*), TLN1 (*Hajagos et al., 2012*), SAG1/ROP1/ROP2-4/ROP7/ROP13/RON9 (gift from Dr J-F Dubremetz), M2AP/proM2AP/CPL (gift from V. B. Carruthers), MIC3 T82C10 (*Achbarou et al., 1991*), MIC6 (*Meissner et al., 2002*), PLP1 (*Roiko and Carruthers, 2013*), MIC4 (*Meissner et al., 2002*), PfSUB1 (a gift from Michael Blackman), Cpn60 (*Agrawal et al., 2009*), RON2/RON4 (gift from M. Lebrun), MIC8 (EGF-N-term) (*Meissner et al., 2002*).

For western blot analyses, secondary peroxidase conjugated goat α-rabbit/mouse antibodies (Sigma) were used. For immunofluorescence analyses, the secondary antibodies Alexa Fluor 488- and Alexa Fluor 594-conjugated goat α-mouse/rabbit antibodies (Life Technologies) were used.

## Parasite transfection and selection

Transfection of *T. gondii* tachyzoites was performed as previously described (*Soldati and Boothroyd, 1993*). The transgenic parasites were selected with mycophenolic acid and xanthine for HXGPRT selection (*Donald and Roos, 1993*), pyrimethamine for DHFR selection (*Donald and Roos, 1993*) or phleomycin for Ble selection (*Messina et al., 1995*). Clones for all stable expressing strains were obtained by limited dilution and checked for genomic integration by PCR and analyzed by IFA and/or WB.

## Preparation of *T. gondii* genomic DNA and RNA

Extraction of genomic DNA (gDNA) from *T. gondii* RH or RHΔKu80 was done with the Wizard SV genomic DNA purification system (Promega).

## DNA vector constructs and transfection

The plasmids for generating endogenously Ty tagged strains of ASP3 was made by amplifying its C-terminal region with primers 5000/5001 (*Supplementary file 6*) and cloning into pG152-KI-3Ty-lox-SAG1_3'UTR-HX (*Pieperhoff et al., 2015*) between KpnI and NsiI (*Supplementary file 6*) to produce the KI-ASP3ty-HX plasmid (*Supplementary file 7*).

The plasmids for generating endogenously tagged strains of other genes were made in a similar fashion with appropriate primers and vectors (*Supplementary files 6–8*). 40 ug of linearized plasmids was transfected in the corresponding strains.

KI-ASP3ty-DHFR plasmid was generated by replacing the HX cassette with DHFR cassette between SacII sites and KI-ASP3myc-Bleo plasmid was generated by subcloning the KI-ASP3 fragment form KI-ASP3ty-HX plasmid into pT8-GRA7myc-Bleo (*Hammoudi et al., 2015*) between KpnI-NsiI sites (*Supplementary file 7*).

To generate the ASP3-iKD strain, a Cas9-YFP/CRISPR gRNA targeting the 5' region of ASP3 was generated using the Q5 site directed mutagenesis kit (NEB) with primers 5040/4883 and using the vector pSAG1::CAS9-GFP-U6::sgUPRT as template (*Shen et al., 2014a*). 5' and 3' homology regions were amplified with primers 5415/5416 and 5417/5418, respectively, and cloned into the 5'COR-pT8TATi1-HXtet07S1mycNtCOR plasmid (*Salamun et al., 2014*) using the sites NcoI-BamHI and AvrII-NotI respectively. 40 ug of this plasmid was digested with Sph-SbfI and co-transfected with 20 ug of the ASP3-Cas9-YFP/CRISPR plasmid into RHΔKu80. Transfected parasites were cloned by FACS sorting the green fluorescent parasites into 96 w plates 48 hr post-transfection. Integration of the inducible cassette was confirmed by PCR of genomic DNA using primers listed (*Supplementary file 6*). ASP3 was endogenously tagged in this background, as described above, using the KI-ASP3ty-DHFR and KI-ASP3myc-Bleo plasmids to generate the ASP3ty-iKD and ASP3myc-iKD strains (*Supplementary file 8*).

The catalytically dead mutant plasmid, pTub8-ASP3ty-D299A-HX, was generated from the pTub8-ASP3ty-HX (*Shea et al., 2007*) using the Q5 site directed mutagenesis kit (NEB) with primers 5735/5736.

For complementation, Ty-tagged full length cDNA of ASP3 and its dead mutant, from pTub8-ASP3ty-HX and pTub8-asp3ty-D299A-HX, were first cloned into a plasmid having UPRT homology arms in both 5′ and 3′ between ApaI-PacI sites to generate 5′UPRT-pT8-ASP3ty −3′UPRT and 5′UPRT-pT8-asp3ty-D299A-3′UPRT plasmids. These plasmids were then co-transfected with 5 ug of the UPRT-Cas9-YFP/CRISPR plasmid into the ASP3myc-iKD strain followed by FACS sorting the green fluorescent parasites into 96 w plates 48 hr post-transfection. The integration of the cassette was confirmed by PCR analysis on genomic DNA using primers 5994/5997.

Plasmid SUB1-Cas9-YFP/CRISPR for inserting a Ty tag before the GPI anchor of SUB1 was generated similarly to the ASP3-Cas9-YFP/CRISPR plasmid with primers 5915/4883 and transfected into the ASP3myc-iKD strain along with 40 ug of a 130 bp double stranded oligos (5916/5917) encompassing the Ty tag.

In order to make the SUB2 knockout strain, part of SUB2 was replaced with a CAT cassette. We first generated two gRNA SUB2-Cas9-YFP/CRISPR plasmids as presented (*Figure 4—figure supplement 4*) using primers 6435/4883 and 6625/4883. Mutagenesis was done as previously described. The second gRNA was inserted into the first plasmid between KpnI-XhoI sites to generate the SUB2-6435-6625-Cas9-YFP/CRISPR plasmid. CAT selection cassette was amplified by KOD DNA polymerase with 28 bp SUB2 homology arms in both 5′ and 3′ with primers 6626/6627 and transfected with 20 ug of the 2-guides SUB2-6435-6625-Cas9-YFP/CRISPR plasmid in the endogenously tagged SUB2ty strain.

The plasmid for recombinant expression of MIC6 was generated by amplifying the first three EGF domains with primers 6283/6285 and cloning it into a pET-GST vector between EcoRI-SpeI restriction sites.

Transient transfection of GRASP-YFP, pTub-ASP3-3Ty and pLIC-PGRA16-GRA16-3Myc (*Hammoudi et al., 2015*) was performed by using 40 µg of each plasmid as previously described (*Soldati and Boothroyd, 1993*).

All primers, plasmids and strains used in this study are listed in *Supplementary files 6*, *7* and *8* respectively.

## Immunofluorescence analysis (IFA)

HFF monolayers on coverslips were infected with *T. gondii* tachyzoites and grown 24–30 hr at 37℃. The coverslips were subsequently fixed with either cold methanol or 4% paraformaldehyde (PFA)/ 0.001% glutaraldehyde (GA) for 10 min, prior to quenching with 0.1 M glycine/PBS. Cells were permeabilized with 0.2% Triton/PBS and blocked with 2%BSA/0.2%Triton/PBS. Following this, cells were probed with primary antibodies diluted in 2%BSA/0.2%Triton/PBS for 1 hr followed by 3 washes with 0.2%Triton/PBS. Cells were then incubated with secondary antibodies in 2%BSA/0.2% Triton/PBS for 1 hr. Parasite and HFF nuclei were stained with DAPI (4′,6-diamidino-2-phenylindole; 50 µg/ml in PBS) and coverslips were mounted on Fluoromount G (Southern Biotech) on glass slides and stored at 4℃ in the dark. Images were recorded on the LSM700 confocal microscope (Zeiss) at the Bioimaging core facility of the Faculty of Medicine, University of Geneva. Final image analysis and processing was done with ImageJ.

## Western blotting

Freshly egressed parasites were pelleted by centrifugation, washed in PBS, and subjected to SDS-PAGE under reducing conditions. Proteins were transferred to nitrocellulose membrane and immunoblot analysis was performed. Primary and secondary antibodies are diluted in 5%milk/0.05% Tween/PBS, washes are performed in 0.05%Tween/PBS.

## Plaque assays

Freshly egressed parasites were inoculated on a confluent monolayer of HFFs and grown for 7 with or without anhydrotetracycline (ATc). Parasites were fixed with PFA/GA followed by staining with Crystal Violet as previously described.

## Invasion assay

Freshly egressed parasites pre-treated 48 hr or 24 hr +/−ATc were inoculated on coverslips seeded with HFF monolayers and centrifuged at 1100 g for 1 min. For invasion assay with 49c, parasites were pre-treated with DMSO/49c (1 μM) for varied time points prior to seeding them for the assay. Invasion was allowed to take place for 20 min at 37°C +/−ATc prior to fixation using PFA/Glu. Extracellular parasites were stained first using monoclonal anti-SAG1 antibody in non-permeabilized conditions. After 3 washes with PBS, cells were fixed with 1% formaldehyde/PBS for 7 min and washed once with PBS. This is followed by permeabilization with 0.2%Triton/PBS and staining of all parasites with polyclonal anti-GAP45 antibody. Appropriate secondary antibodies were used as previously described. 100 parasites were counted for each experiment, and the ratio between red (all) and green (invaded) parasites is presented. Results are mean ±standard deviation of three independent biological replicate experiments.

## Induced egress assay

Freshly egressed parasites, pre-treated 24 hr +/−ATc, were inoculated on coverslips with HFF monolayer and grown for 30 hr +/−ATc at 37°C. For egress assay with 49c, parasites were pre-treated with DMSO/49c for 12 hr prior to plating freshly egressed tachyzoites to a new monolayer of HFF and grown for 30 hr and treatment with DMSO or 49c (1 μM) were done for varying time points. In some cases 49c is washed away 3 hr or 6 hr before induced egress.

Following a serum-free DMEM wash, the infected HFF monolayers were incubated with 3 μM of the Ca2+ ionophore A23187 (from *Streptomyces chartreusensis*, Calbiochem) in serum-free DMEM for 7 min at 37°C. Cells were fixed with PFA/Glu, and processed for IFA with anti-GAP45 antibody. 100 vacuoles were counted per strain and scored as egressed or non-egressed. Results are mean ±standard deviation of three independent biological replicate experiments. Control experiment with DMSO showed no egress.

For live video microscopy of induced egress, parasites were grown on 5 mm Fluorodishes (World Precision Instruments) seeded with HFF monolayers for ~30 hr at 37°C and egress was induced as described above.

## Microneme secretion assay

Microneme secretion assay was performed on freshly egressed parasites, with or without 24 hr/48 hr ATc treatment or DMSO/49c (1 μM). Parasites were resuspended in equal volume of intracellular (IC) buffer (5 mM NaCl, 142 mM KCl, 1 mM MgCl2, 2 mM EGTA, 5.6 mM glucose, 25 mM HEPES, pH to 7.2 with KOH) and pelleted at 1050 rpm for 5 min. The pellets were washed again in IC buffer before resuspension in 100 μl serum-free media with 2% ethanol for 30 min at 37°C. Following this incubation, parasites were pelleted at 1000 g for 5 min at 4°C, and the supernatant was transferred to new Eppendorf tubes (the pellet from this step serves as the pellet fraction) and re-pelleted at 2000 g for 5 min at 4°C. The final supernatant, containing the excreted secreted antigens (ESA), and pellet fractions were resuspended in SDS loading buffer and boiled prior to immunoblotting.

## Evacuole formation assay

Rhoptry content secretion was assayed by means of the evacuole detection assay, as previously described (*Håkansson et al., 2001*; *Kessler et al., 2008*). Freshly egressed extracellular parasites were incubated Endobuffer (44.7 mM K2SO4, 10 mM MgSO4, 106 mM sucrose, 5 mM glucose, 20 mM Tris–H2SO4, 3.5 mg/ml BSA, pH 8.2) containing ±1 μM Cytochalasin D for 10 min at room temperature. The parasites were then added to HFF-coated coverslips, allowed to settle and incubated at 37°C for 15 min. The medium was replaced with complete DMEM medium containing ±1 uM Cytochalasin D and incubated at 37°C for an additional 20 min followed by fixation with 4% paraformaldehyde for 10 min. IFA was performed as described before. ROP1 is used as a rhoptry marker for the visualisation of the evacuoles, while GAP45 was used to stain the parasites.

For analysis of MJ with RON4, 20 min permeabilization with 0.1% saponin was performed. Both primary and secondary antibody incubations to distinguish secreted RON4 from intracellular RON4 were performed in 2%BSA/PBS. The parasite's pellicle was stained with GAP45 antibody. 200 parasites were analyzed for RON4 secretion per experiment and quantified for the presence of the apical

tip staining. The results are mean ±standard deviation of three independent biological replicate experiments.

## Intracellular growth assay

Freshly egressed parasites, pre-treated 24 hr +/−ATc, were allowed to grow on HFF monolayers with +/−ATc for 24 hr prior to fixation with PFA/GA. For intracellular growth assay, wild type prasites were seeded on HFF monolayers and were allowed to grow in the presence of DMSO/49c (0.5 µM, 1 µM, 5 µM) for 24 hr prior to fixation. IFAs were done using α-GAP45 antibodies to detect parasites. The number of parasites per vacuole was scored, counting 200 vacuoles for each condition. Results are mean ±standard deviation of three independent experiments.

## Gliding motility assay

Freshly egressed parasites pre-treated 48 hr +/−ATc or DMSO/49c (1 µM) were pelleted and resuspended in serum free DMEM before adding to poly-Lysine coated coverslips in a 24-well plate. The plate was centrifuged for 1 min at 1100 g to settle the parasites onto the coverslips. The media is aspirated and replaced with serum-free media containing DMSO or 3 µM A23187. Following incubation for 30 min at 37°C, the coverslips are fixed with PFA/Glu. The deposited trails were visualized by non-permeabilizing IFA using anti-SAG1 Abs. Results presented are representative of three independent experiments.

## Host cell attachment assay

Attachment to HFF monolayers was assessed as previously described (*Mueller et al., 2013*). Freshly egressed GFP expressing WT parasites and ASP3-myc-iKD parasites (+/-ATc 48 hr or 24 hr) were mixed in a 1:1 ratio in Endobuffer (44.7 mM K2SO4, 10 mM Mg2SO4, 106 mM sucrose, 5 mM glucose, 20 mM Tris, 0.35% wt/vol BSA, pH 8.2) containing 1 µM cytochalasin D (Sigma-Aldrich). Following 10 min incubation at room temperature, parasites were added to a HFF-coated coverslip (assay) or Poly-L-lysine (control) and centrifuged for 1 min at 1000 rpm. Control samples were immediately fixed with PFA/GA for 7 min at RT. For the assay samples, medium was replaced with pre-warmed DMEM 5% FCS containing 1 µM cytochalasin D to prevent invasion, incubated for 15 min at 37°C and then fixed with PFA/GA for 7 min. Immunofluorescence was performed using α-GAP45 and Alexa647 ('red') as secondary antibodies. Ratio between red/green (all) and red only (ASP3-myc-iKD) attached parasites was counted. 100 parasites were screened for each condition. Results are mean ±standard deviation of three independent experiments.

## Transmission electron microscopy

Freshly egressed parasites, pre-treated 24 hr +/−ATc, were allowed to grow on HFF monolayers for 24 hr, either in the presence of 1 µM 49c or +/−ATc, prior to fixation. Infected HFFs were washed with 0.1M phosphate buffer pH 7.2 and subsequently fixed with 2.5% glutaraldehyde in 0.1 M phosphate buffer pH 7.2. This is followed by post-fixation in osmium tetroxide, dehydration in ethanol and treatment with propylene oxide before embedding in Spurr's epoxy resin. Thin sections were stained with uranyl actetate and lead citrate prior to examination using a Technai 20 electron microscope (FEI Company). Three independent samples were prepared for each condition and multiple thin sections were analyzed for each sample.

## Protease substrate screen using terminal amine isotopic labelling of substrates (TAILS)

ASP3 candidate substrates were identified using the methodology previously described (*Kleifeld et al., 2010*; *Kleifeld et al., 2011*). TAILS uses primary amine labelling-based quantification as the discriminating factor, negatively selecting non-natural N-terminal peptides by using depletion with a dendritic polyglycerol aldehyde polymer (HPG-ALD). In addition, besides substrate protein identification, TAILS allows mapping of cleavage sites with amino acid precision. The TAILS procedure involves, tryptic digestion of the TMT labelled sample proteome (ASP3ty-iKD parasites +ATc/-ATc), which generates internal tryptic peptides with free amines at N termini (Pre-TAILS sample). The tryptic peptides were depleted by amine-reactive aldehyde-derivatized polymer. The unbound peptides (TAILS sample) consist of highly enriched isotopically labelled natural and *T. gondii*

processing protease-derived N-terminal peptides. This fraction is then analyzed and quantified by high-accuracy LC-MS/MS.

## Sample preparation

In brief, *T. gondi* protein extract was generated by treatment of cells with RIPA buffer (150 mM NaCl, 1% NP40, 0.5% natriumdeoxycholate, 0.1% SDS, 50 mM Tris-HCl, pH 8.0, 1 mM EDTA) for 30 min at 4°C, high-speed centrifugation for 15 min, and subsequent acetone precipitation. The acetone pellet was dissolved completely in 8M GuHCl. The sample was adjusted to 2.5 M GuHCl, 100 mM HEPES, pH 7.8 (TAILS buffer) at 2 mg/ml protein and denatured at 65°C for 15 min. The sample was reduced for 30 min at 65°C in the dark by adding 0.01 volumes of 350 mM Tris (2-carboxyethyl) phosphine (TCEP), and alkylated by addition of 0.02 volumes of 250 mM iodoacetic acid (IAA) and incubation for 30 min at room temperature. For isobaric labeling of protein N-termini we applied TMT10plex reagents (Thermo Scientific) at 0.8 ug/100 ug of denatured protein in 50% DMSO. A labeling scheme is provided in the supplementary materials (*Figure 3—figure supplement 1*). Subsequent to access label quenching by $(NH_4)_2CO_3$ addition (100 mM final) and reagent cleanup by acetone-methanol precipitation, we digested proteins to peptides by adding trypsin at a 100:1 substrate/enzyme ratio (w/w). Following overnight digestion, peptides having a free alpha amine were removed from the sample by HPG-ALD polymer treatment in the presence of 20 mM $NaBH_3CN$. Prior to HPG-ALD polymer pullout (po) we removed 10% of the sample for separate analysis by LC-MS (termed Pre-TAILS sample). Finally, we separated free peptides from polymer by ultra-filtration (Amicon ultra 30 kDa, Millipore) and purified TAILS and Pre-TAILS peptides by reverse phase chromatography on C18 SPEs (Waters).

## LC-MS analysis

Peptides were analyzed on an Orbitrap Fusion Tribrid Mass Spectrometer (Thermo Scientific) operated in line with an EASY-nLC 1000 Liquid Chromatography system (Thermo Scientific) in data dependent analysis mode (DDA). In brief, we separated peptides by reverse phase (RP) chromatography on custom made 150 cm x 75 µm ID frit columns packed with Reprosil-Pur 120 C18-AQ, 1.9 µm (Dr. Maisch GmbH). The liquid phase was mixed from 0.1% FA in water (A) and 0.1% FA in ACN (B) both obtained in MS-grade from Biosolve. For separation, peptides were directly loaded onto the analytical column and resolved by applying a linear gradient from 2% to 35% B in 120 min at a constant flow rate of 300 nl/min. The analytical column was heated to 50°C by an electronic column heater for fused silica columns (PST). Eluted peptides were electrosprayed into the mass spectrometer using a 10 µm PicoTip Emitter (New Objective) in combination with a Nanospray Flex Ion Source (Thermo Scientific) operated at a spray voltage of 2.6 kV. The MS method was composed of a precursor scan (MS1) from 350 to 1250 m/z at an orbitrap resolution of 60 k (maxIT: 60 ms, AGC target: 5e5) followed by fragment ion scans (MS2) acquired post HCD activation at a collision energy of 35% in the orbitrap at a resolution of 60 k (maxIT: 118 ms, AGC target: 1e5). The first mass for all MS2 scans was set to 100 m/z and the cycle time was limited to 2 s (top speed option). Precursors were selected by quadrupole isolation using a window of 1.6 m/z around the respective target ion. Target ions and their respective isotopes were excluded from fragmentation for 20 s after the 2nd appearance within 15 s. In general, only ions of the charge state 2–5 were considered for fragment ion spectrum recording. We preformed LC-MS data analysis using proteome discoverer 2.1 (Thermo Scientific). In brief, we imported all LC-MS runs as fractions ignoring MS peaks below 3 S/N. Subsequently, we matched MS2 spectra to the *Toxoplasma gondi* proteome (5811 GT1) using the SEQUEST HT search engine and the following settings: Enzyme specificity: semi-ArgC, Peptide mass tolerance: 10 ppm, Fragment mass tolerance: 0.05 Da, Ion series: y- and b incl. neutral losses; Static modifications: Carbamidomethyl (C) and TMT6plex (K); Variable modifications: Oxidation (M), Acetylation or TMT6plex (peptide n-terminus), Met-loss (protein terminus). FDR was set to 1% by the target +decoy approach in conjunction with Percolator (*Käll et al., 2007*). Reporter ions were extracted with an integration tolerance of 20 ppm. The resulting reporter ion abundance (S/N) was normalized using the total peptide approach implemented in pd 2.1 (median normalization) and summarized on the peptide group level applying a co-isolation threshold of 30% per matched PSM. For further statistical analysis, we imported the Peptide Group report into the R software environment for statistical computing (version 3.3.2).

## Parsing of datasets and interpretation

Generally speaking, abundance ratios of <1 were interpreted as proteolytic processing due to active ASP3 in –ATc conditions, that is, higher abundance of the correctly processed (acetylated) N-terminal peptide. Conversely, ratios >1 may be either due to higher abundance of peptides representing unprocessed N-termini or accumulation of degradation products in +ATc conditions. However, for the purpose of identifying likely biologically relevant changes of +ATc/-ATc peptide ratios we calculated stringent cut-offs based on the distribution of log2 ratios in the normalized pre-TAILS dataset. Using an assumption-free empirical cumulative distribution function (ECDF) to determine the distribution of log2 ratios for R.XXXXR peptides, that is, all peptides generated by tryptic cleavage (pre-TAILS data) cut-off ratios at the 1% and 99% quantiles were calculated. The calculated log2 values of −0.4719 (0.72) and 0.919 (1.89), respectively, predict 1% false-positives in the dataset. However, manual surveillance of the data indicated that a more stringent lower cut-off ratio of 0.22 was more appropriate considering that ASP3 is virtually absent in +ATc conditions. Instead of the calculated upper cut-off (log2 0.9119) we applied a slightly more stringent threshold of 2-fold higher abundance of peptides in +ATc conditions as secondary criterion for interpretation of the TAILS dataset in terms of ASP3 substrate specificity. In addition, we also considered more relaxed selection criteria with +ATc/-ATc ratio cut-off above or below 2-fold change on either side, i.e. ratios of <0.5 and>2, respectively). The criteria for substrate selection and further pertinent analysis are described in Appendix 1.

The LC-MS-MS datasets generated in this study have been deposited on ProteomeXchange Consortium via the PRIDE (*Vizcaíno et al., 2016*) partner repository with the dataset identifier PXD006235.

## In vitro measurement of IC50

*T. gondii* tachyzoites expressing luciferase (RH-pTub-CBG99-luciferase) (200 parasites) were added in HFF coated 96-well plate harbouring $10^5$ cells per well. Inhibitors, (49c and 49b) along with Pyrimethamine (used as a positive control in this experiment, *Donald and Roos, 1993*), were diluted in supplemented DMEM and added to the monolayers at various concentrations in triplicates along with DMSO treated control for each set. The assay was performed in a 100 µl final volume. The plates were incubated for 96 hr at 37°C, cells were washed once with PBS and lysis was performed in the wells with 100 µl lysis buffer (20 mM Tris HCl pH 7.5, 10% Glycerol, 1% Triton X-100, 2 mM DTT). Uninfected host cells were also lysed as negative control. 20 µl of lysate was added to 96well MicroliteTM TCT plate and incubated with 20 µL of Luciferase substrate solution (1 mM Luciferin, 3 mM ATP, 15 mM MgSO4.7H2O, 30 mM Tris HCL pH7.5) and reading was taken immediately. Growth of *T. gondii* was evaluated by measurement of relative luciferase unit (RLU) in the SynergyH1 multi-well plate reader (BioTek). Assays were performed in triplicates for each drug and IC50 (50% reduction in luciferase activity as compared no drug treated control) were determined by plotting luminescence against the drug concentrations from analysis of dose response curve using Prism 5.0 (GraphPad, San Diego, CA).

## In vitro cleavage assays

Protease cleavage assays were performed using ASP3ty or with the catalytically dead asp3ty-D299A immuno-purified from parasite lysates using mouse αTy antibody, as described previously (*Boddey et al., 2010*; *Coffey et al., 2015*). Proteases bound to agarose was prepared by incubating αTy-agarose in parasite lysate, for 2 hr at 4°C, followed by extensive wash in 1% TritonX-100/PBS and stored in PBS. Either 40 ul of ASP3ty or asp3ty-D299A -agarose-bound proteins were resuspended in digestion buffer (25 mM Tris.HCl, 25 mM MES, pH 5.5; different pH ranges were tested and pH 5.5 was optimal) with 20 uM of synthetic peptide substrate (ThermoFisher scientific, >98% purity) MIC6: DABCYL-G-FVQLSETPAA-G-EDANS in 100 µL of total volume. Samples were gently shaken during incubation to disperse protease-agarose. Samples were incubated at 37°C for 5 hr and processing measured using a SynergyH1 multi-well plate reader (BioTek) excited at 340 nm and reading emissions at 490 nm. Change in Relative Fluorescence Unit (ΔRFU) was measured for each time point (0, 60, 120, 180, 240, 300 mins) by subtracting RFU from blank (without enzyme) for each time point. Inhibition of ASP3 by compounds 49b (1 µM), 49c (100 nM) and pepstatin (10 µM) was performed in exact similar manner as mentioned above by performing ASP3 cleavage assay (ASP3ty)

in presence of each compound. The results were plotted using Prism 5.0 (GraphPad, San Diego, CA). Cleavage assay with recombinant MIC6 was performed with approximately 1 µg of recombinant MIC6. Recombinant MIC6 was incubated with immunoprecipitated ASP3ty or asp3ty-D299A resuspended in digestion buffer (50 µL total volume) for 5 hr at 37°C with constant shaking. In some case the assay was performed in presence of inhibitors such as Pepstatin (10 µM), 49c (1 µM) or 49b (1 µM). Samples were resuspended in SDS loading buffer and analyzed by WB with α-Profilin-GST and α-Ty Ab.

## Expression of recombinant MIC6

Recombinant MIC6 was expressed in *E.coli* harbouring plasmid pET-GST-MIC6, which encodes residues 15–288 of MIC6 (containing amino acid sequence $_{91}$VQLSETPA$_{98}$) fused with GST in the N-terminal.

## Phylogenetic tree construction

Sequences of Apicomplexan aspartyl proteases were procured from EuPathDB and aligned using MUSCLE sequence alignment software (*Edgar, 2004a*, *2004b*). The resulting sequence alignment was manually curated utilizing BioEdit (http://www.mbio.ncsu.edu/bioedit/bioedit.html) to edit out uninformative alignment positions. Phylogeny tree was generating utilizing PhyML (*Guindon et al., 2010*) on the curated MUSCLE alignment, using LG model of amino acids substitution with NNI topology search. Phylogeny.fr (*Dereeper et al., 2008*) platform was utilized for much of the above analysis. The EupathDB IDs of sequences used in the phylogenetic analysis were listed in *Supplementary file 9*, and the curated alignments shown in *Supplementary file 10*.

## Acknowledgements

We would like to thank Valérie Polonais for her contribution and initial investigation into aspartyl proteases in *Toxoplasma gondii*. This work was supported by Carigest SA, the Swiss National Science Foundation (FN3100A0-116722 to DS-F and CRSII3_16002 to AH and DSF). DSF is an HHMI senior international research scholar. BM is recipient of long-term EMBO fellowships. SKD was supported from the Sir Jules Thorn Charitable Overseas Trust reg., Schaan (Dr. Karine Frenal) and MalarX (SystemsX.ch).

## Additional information

### Competing interests

Dominique Soldati-Favre: Reviewing editor, *eLife*. The other authors declare that no competing interests exist.

### Funding

| Funder | Grant reference number | Author |
|---|---|---|
| Carigest SA | | Damien Jacot |
| Swiss National Science Foundation | 310030B_166678 | Sunil Kumar Dogga |
| EMBO long-term fellowship | | Budhaditya Mukherjee |
| Indo-Swiss joint research programme | | Dominique Soldati-Favre |
| Swiss National Foundation | Sinergia | Adrian B Hehl |

The funders had no role in study design, data collection and interpretation, or the decision to submit the work for publication.

## Author contributions
Sunil Kumar Dogga, Conceptualization, Data curation, Formal analysis, Validation, Investigation, Visualization, Methodology, Writing—original draft; Budhaditya Mukherjee, Data curation, Formal analysis, Investigation; Damien Jacot, Data curation, Formal analysis, Investigation, Writing—review and editing; Tobias Kockmann, Resources, Investigation, Methodology, Writing—original draft; Luca Molino, Data curation, Investigation; Pierre-Mehdi Hammoudi, Ruben C Hartkoorn, Formal analysis, Investigation; Adrian B Hehl, Data curation, Formal analysis, Writing—original draft; Dominique Soldati-Favre, Conceptualization, Formal analysis, Supervision, Funding acquisition, Project administration, Writing—review and editing

## Author ORCIDs
Sunil Kumar Dogga (iD) https://orcid.org/0000-0001-9158-9612
Budhaditya Mukherjee (iD) https://orcid.org/0000-0002-1058-3620
Dominique Soldati-Favre (iD) https://orcid.org/0000-0003-4156-2109

## Decision letter and Author response
Decision letter https://doi.org/10.7554/eLife.27480.049
Author response https://doi.org/10.7554/eLife.27480.050

---

# Additional files

## Supplementary files
• Supplementary file 1. Proteins represented by peptides with normalized +ATc/-ATc abundance ratios of <0.22 (stringent). Predicted or experimentally validated subcellular localization is indicated. Gene IDs refer to ToxoDB, Release 31.
DOI: https://doi.org/10.7554/eLife.27480.034

• Supplementary file 2. Proteins represented by peptides with normalized +ATc/-ATc abundance ratios of <0.5 (relaxed). Predicted or experimentally validated subcellular localization is indicated. Gene IDs refer to ToxoDB, Release 31.
DOI: https://doi.org/10.7554/eLife.27480.035

• Supplementary file 3. Proteins represented by peptides with normalized +ATc/-ATc abundance ratios of >2. Predicted or experimentally validated subcellular localization is indicated. Gene IDs refer to ToxoDB, Release 31.
DOI: https://doi.org/10.7554/eLife.27480.036

• Supplementary file 4. Eight proteins represented by peptides with normalized +ATc/-ATc abundance ratios of <0.22 and >2 (see also *Figure 3C*). Predicted or experimentally validated subcellular localization is indicated. Gene IDs refer to ToxoDB, Release 31.
DOI: https://doi.org/10.7554/eLife.27480.037

• Supplementary file 5. Detailed analysis of MIC3-derived peptide abundances and +ATc/-ATc abundance ratios (ProteomeDiscoverer output). Color code: ratios > 2, pale red rows; ratios < 0.22, pale green rows; abundance ratios, yellow column (AC); normalized peptide abundances (+ATc), green columns (AO-AR); normalized peptide abundances (-ATc), purple columns (AS-AV); highly abundant peptide representing the ASP3-processed mature MIC3 N-terminus, red fields (32 AO-AR).
DOI: https://doi.org/10.7554/eLife.27480.038

• Supplementary file 6. List of oligonucleotide primers used in this study.
DOI: https://doi.org/10.7554/eLife.27480.039

• Supplementary file 7. List of plasmids generated for this study
DOI: https://doi.org/10.7554/eLife.27480.040

• Supplementary file 8. List of strains generated for this study
DOI: https://doi.org/10.7554/eLife.27480.041

• Supplementary file 9. List of EuPathDB IDs of protein sequences used for phylogeny tree generation.
DOI: https://doi.org/10.7554/eLife.27480.042

- Supplementary file 10. Curated alignments used to generate the phylogenetic tree of Apicomplexan aspartyl proteases
DOI: https://doi.org/10.7554/eLife.27480.043
- Transparent reporting form
DOI: https://doi.org/10.7554/eLife.27480.044

### Major datasets

The following dataset was generated:

| Author(s) | Year | Dataset title | Dataset URL | Database, license, and accessibility information |
|---|---|---|---|---|
| Dogga SK, Mukherjee B, Jacot D, Kockmann T, Molino L, Pierre-Mehdi Hammoudi, Hartkoorn RC, Hehl AB, Soldati-Favre D | 2017 | Toxoplasma gondii aspartyl protease 3 | http://www.ebi.ac.uk/pride/archive/projects/PXD006235 | The mass spectrometry proteomics data have been deposited to the ProteomeXchange Consortium via the PRIDE partner repository with the dataset identifier PXD006235 |

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

# Appendix 1

DOI: https://doi.org/10.7554/eLife.27480.045

As the primary selection criterion for identifying potential ASP3 substrates we considered peptide ratios of <0.22 or <0.5 (relaxed) representing N-termini, which are significantly less abundant in +ATc conditions, i.e. when ASP3 activity is essentially abolished. This is true for 23 (95) peptide groups representing 17 (59) proteins, 10 (29) of which have a confirmed/predicted signal peptide (*Supplementary files 1–3*). Of note, and in line with the invasion and egress phenotypes, half of the predicted secreted proteins in these datasets are annotated in ToxoDB as ROP, RON or MIC proteins, some of which are known to be processed in a post-Golgi compartment.

In addition to lower abundance/disappearance of processed N-termini generated by ASP3 in +ATc conditions as a primary criterion for identifying ASP3 substrates, we also identified 42 peptide groups with +ATc/-ATc ratios >2 representing 29 proteins, 21 of which have a predicted signal sequence (*Supplementary file 3*, *Figure 2C*). Because ASP3 is an essential, post-Golgi resident processing enzyme, we reasoned that parasite growth in +ATc conditions might lead to a general perturbation of their distal secretory system resulting in an increased occurrence of N-terminal peptides representing degradation products. More importantly, this also predicts that unprocessed direct or indirect ASP3 substrates may be subject to an increased rate of degradation eliciting increased proportions of tryptic peptides that are not represented in –ATc conditions. Indeed, by intersecting both datasets, we identified 5 (8) proteins (*Supplementary file 4*, *Figure 2C*), which are represented by peptides with strongly polarized +ATc/-ATc ratios. Using relaxed criteria this set comprises SUB1, MIC3, MIC11, ROP9, ROP16, and ROP40, as well as two hypothetical transmembrane proteins of unknown function and localization. While we consider this as robust support for candidate ASP3 substrates, the 17 remaining secreted proteins in the curated TAILS dataset represented only by peptides with a ratio <0.5 are also considered valid candidate substrates (*Figure 2C*). In addition, considering that N-termini may not be detected in the LC-MS-MS analysis for purely technical reasons (cleavage generates peptides incompatible with LC-MS analysis due to peptide length, charge state or hydrophobicity), this leaves the possibility that some ASP3 substrates are represented only by peptides with a ratio >2, that is, putative degradation products.

The most robust indication for directly ASP3-dependent processing events are +ATc/-ATc ratios of <0.22 (0.5) for N-terminal peptides in conjunction with the detection of additional peptides representing immature or degraded products with ratios >2. While this is the case for some examples such as MIC3 or MIC11, many of the 12 (39) proteins represented only by the former require additional evaluation. Experimentally validated ASP3 substrates provide an excellent framework for interpreting TAILS datasets. MIC3 is a classic case of experimentally validated posttranslational processing represented in TAILS. The MIC3 N-terminal peptide [F]. AVTETHSSVQSPSKQETQLCAISSEGKPCR.[N] (57–869 has an acetyl modification at the N-terminal alanine and an +ATc/-ATc ratio of 0.074 which is 3-fold lower than the stringent cutoff of <0.22. A second, non-acetylated N-terminus at S67 is also represented with a peptide ratio of 0.079. Prima facie the former would be considered the more likely mature N-terminus of the processed ASP3 substrate. However, if normalized abundances are taken into account it becomes clear that the latter is represented with a >100 fold higher abundance (*Supplementary file 5*, Line 35, Columns AO-AR). Although a minor proportion of MIC3 is processed at position S69 there is no indication in the TAILS dataset that this is due to technical reasons and therefore likely represents a natural variability. Taken together, both qualitative and quantitative parameters of the TAILS data are in complete agreement with the predicted cleavage site at [SSVQ | SPSK] and in support of MIC3 being an ASP3 substrate. MIC11 provides another, albeit non-validated example presenting a very similar profile in

TAILS. A highly abundant N-terminal peptide [T].EDDKSAASIVR.[G] (58-68) with a ratio of 0.045 robustly predicts an ASP3-dependent cleavage site in this 204 aa protein at [VETT | EDDK]. Here too, detection of an N-terminal peptide shifted by two amino acids (56-68) with a ~15 fold lower abundance likely represents a minor alternative cleavage site. In addition, four peptides with ratios >2 were detected, one of which is highly abundant in +ATc conditions and likely represents an immature N-terminus after cleavage of the signal peptide.

Some ASP3 substrates may be revealed in the TAILS dataset only by peptides with ratios <0.22 (0.5). This is the case for MIC6 whose predicted ASP3 cleavage site [VQLS | ETPA] is represented by the abundant peptide [S].ETPAACSSNPCGPEAAGTCKETNSGYICR.[C] with an +ATc/-ATc ratio of 0.089 in the TAILS dataset. No peptides with ratios >2 have been detected in this experiment. ASP3-dependent processing of MIC6 has been experimentally validated (*Figure 3B*), as well as for the hypothetical protein TGGT1_273860 (TAILS3) (*Figure 5A*), which presents a very similar profile in TAILS analysis. Conversely, ROP1 was only represented by peptides with +ATc/-ATc ratios >2 in the TAILS dataset. Nevertheless, in vitro cleavage experiments show ASP3-dependent processing producing a mature protein of ~55 kDa (*Figure 4A*). Although two acetylated N-termini were detected in MS analysis, in this case, likely for technical reasons (e.g. flyability), the quantitative information was not sufficient to calculate peptide abundances. However, the high number of peptides (4 N-terminal, 5 internal) in the TAILS profile with ratios >2 is consistent with ROP1 being an ASP3 substrate.

Taken together, TAILS allowed efficient (albeit not comprehensive) identification of ASP3 substrates by interpretation of +ATc/-ATc peptide ratios, and provided biochemical information on processed N-termini.

