## [Decision Letter]

Thank you for submitting your article "A druggable secretory protein maturase of Toxoplasma essential for invasion and egress" for consideration by *eLife*. Your article has been reviewed by two peer reviewers, and the evaluation has been overseen by a Reviewing Editor and Ivan Dikic as the Senior Editor. The following individual involved in review of your submission has agreed to reveal his identity: Mohamed-ali Hakimi (Reviewer #3).

The reviewers have discussed the reviews with one another and the Reviewing Editor has drafted this decision to help you prepare a revised submission.

This is a thoroughly conducted study that defines the role of an important protease called Asp3 in *Toxoplasma gondii*. The data convincingly shows that this protease is necessary for maturation of secretory proteins derived from the rhoptries and micronemes. Conditional knockdown results in a string phenotype in invasion and egress. The authors also demonstrate that a potent inhibitor of ASP3 in vitro, phenocopies the phenotype of loss of ASP3 expression.

Major comments:

In general the authors are miscalling the inhibitor as "selective". The authors are using this to indicate that 49c is potent while 49b is not. This is not selectivity in the typical sense used in inhibitor studies. Selectivity implies that an inhibitor does not act on other targets, such as other plasmepsins, not that it has an activity not seen in related chemical structures. This is not simply a semantic issue as it relates to the point above and raises the question: could 49c act on other proteases in the parasite and might this contribute to its in vivo potency?

In fact, there is no data showing that the hydroxyethylamine compound "selectively" targets ASP3. It is true that this compound inhibits ASP3, this does not mean that it is acting on ASP3 selectively. Given the number of off target effects that have been found for previously described "selective" inhibitors, some caution is due here. Whilst it is clear from the data that the inhibitor blocks ASP3 activity on some substrates in vitro and also phenocopies the conditional knockdown, this does not rule out that an alternative target is partially responsible for its activity in blocking the lytic growth cycle. It would therefore be important to show that non Asp3-targets are not affected by the drug, or ideally identify Asp3 mutant alleles that confer 49c resistance. That could prove difficult but at the very least, the authors should test MIC5 and PLP1 processing, which is Asp3-independent, and thus should not be sensitive to 49c treatment.

In general, it would be less prone to controversy to state: "a potent inhibitor of ASP3 in vitro interrupted the lytic growth cycle, suggesting that targeting ASP3 may be an effective means of blocking parasite growth."

A major finding presented here is that ASP3 is playing an essential role at the invasion step that correlates with the absence of TJ/MJ at the surface of host cells. The authors rule out an effect of asp3 mutant on MIC secretion based on ESA assay and analysis of the two microneme proteins MIC2 and AMA1. Then, the authors are backing up on MIC analysis to reveal that asp3 affects MIC processing. An excellent candidate that could explain the phenotype of asp3 on at the parasite invasion step is MIC8. However, this hypothesis is ruled out based on the argument that MIC8 activity is not dependent on a processing step for maturation. Given the phenotype of asp3 on the TJ/MJ, notably RON4 whose processing is not affected by asp3, it sounds reasonable to think that asp3 depletion could have some effect on secreted proteins although their processing remains independent of ASP3. Is there any assay that could determine whether MIC8 is still secreted in the absence of ASP3?

The authors report the localization of Asp3 and tried to validate its localisation in Golgi and endosomal like compartments. The protein was however not found residing in these compartments (Figure 1). Rather it is expected to localize to post-Golgi compartment. Ectopically expressed and the WT Asp3 have been reported to localise in this sub-compartment by Shea et al., 2007, which authors have acknowledged. In this case for further proof, experiment using brefeldin A (PMID: 1629235, PMID: 11532123) is advisable. By inhibition of trans Golgi transport, the substrates of Asp3 are expected to accumulate in the post-Golgi region. And, if authors can use any known marker of post-Golgi compartments, the localisation of Asp3 will be more clear.

---

## [Author Response]

*Major comments:*

*In general the authors are miscalling the inhibitor as "selective". The authors are using this to indicate that 49c is potent while 49b is not. This is not selectivity in the typical sense used in inhibitor studies. Selectivity implies that an inhibitor does not act on other targets, such as other plasmepsins, not that it has an activity not seen in related chemical structures. This is not simply a semantic issue as it relates to the point above and raises the question: could 49c act on other proteases in the parasite and might this contribute to its* in vivo *potency?*

We fully concur with the reviewers that our statement was misleading given that the data presented in this study ONLY refer to the potency and not to the selectivity of 49c.

In *Toxoplasma gondii* tachyzoites, the other expressed ASPs proteases are dispensable. RH strain parasites individually lacking TgASP1, TgASP2 are readily propagating in culture and are virulent in mice while TgASP5 is dispensable but shows a significant decrease in fitness and virulence (Curt-Varesano et al., 2015; Hammoudi et al., 2015; Coffey et al., 2015). TgASP4, TgASP6 and TgASP7 are not expressed in tachyzoites. In consequence, the fact that 49c phenocopies depletion of ASP3 is not a solid evidence for selectivity.

Since TgASP5 plays a key role in the formation of the intravacuolar membranous nanotubular network (MNN), we examined parasites depleted in ASP3 and parasites treated with 49c by EM and showed that MNN formation is not affected, inferring that 49c does not inhibit TgASP5. In addition, we add data pertaining to the action of 49c on ASP5, showing that 49c has no impact on maturation of ASP5 or on the processing and export of GRA16 (an ASP5 substrate).These data are now included in the new Figure 6—figure supplement 2.

*In fact, there is no data showing that the hydroxyethylamine compound "selectively" targets ASP3. It is true that this compound inhibits ASP3, this does not mean that it is acting on ASP3 selectively. Given the number of off target effects that have been found for previously described "selective" inhibitors, some caution is due here. Whilst it is clear from the data that the inhibitor blocks ASP3 activity on some substrates* in vitro *and also phenocopies the conditional knockdown, this does not rule out that an alternative target is partially responsible for its activity in blocking the lytic growth cycle. It would therefore be important to show that non Asp3-targets are not affected by the drug, or ideally identify Asp3 mutant alleles that confer 49c resistance. That could prove difficult but at the very least, the authors should test MIC5 and PLP1 processing, which is Asp3-independent, and thus should not be sensitive to 49c treatment.*

As recommended, we have examined the processing of TgMIC5 in parasites treated with 49c. As observed upon depletion of TgASP3, TgMIC5 is still processed in presence of 49c. The data is presented in Figure 6—figure supplement 1. In contrast PLP1 is not a suitable control since our results indicate that this protein is affected/processed by ASP3. The analysis of PLP1 processing has been added to Figure 6—figure supplement 1 as well.

As rightly stated by the reviewers, the generation of an *asp3* mutant allele that confers 49c resistance would provide an unambiguous evidence for 49c selectively toward ASP3. We have addressed the selectivity of 49c toward the cluster of aspartyl proteases that groups TgASP3 and the Plasmepsins IX and X in a separate study. The work is based on 3D modeling of these enzymes and docking of substrates as well as the peptidomimetic inhibitor 49c. The results led to a plausible molecular explanation for the selective sensitivity of TgASP3, PMIX and PMX to 49c compared to the other apicomplexan ASPs and Plasmepsins. Importantly were able to confirm the prediction experimentally. We also succeeded in generating independently and reproducibly *T. gondii* parasites mutant resistant to 49c in the ASP3 allele. In our view, this study is beyond the scope of this work and to be submitted for consideration for publication elsewhere.

*In general, it would be less prone to controversy to state: "a potent inhibitor of ASP3* in vitro *interrupted the lytic growth cycle, suggesting that targeting ASP3 may be an effective means of blocking parasite growth."*

We have shown that 49c does not recapitulate any of the ASP5-KO phenotypes and that 49c does not process TgMIC5. Consistently with what is presented here and in the absence of an 49c resistant ASP3 mutant allele, we have rephrased ours findings as recommended, by toning down the claim about the selectivity of 49c toward ASP3.

*A major finding presented here is that ASP3 is playing an essential role at the invasion step that correlates with the absence of TJ/MJ at the surface of host cells. The authors rule out an effect of asp3 mutant on MIC secretion based on ESA assay and analysis of the two microneme proteins MIC2 and AMA1. Then, the authors are backing up on MIC analysis to reveal that asp3 affects MIC processing. An excellent candidate that could explain the phenotype of asp3 on at the parasite invasion step is MIC8. However, this hypothesis is ruled out based on the argument that MIC8 activity is not dependent on a processing step for maturation. Given the phenotype of asp3 on the TJ/MJ, notably RON4 whose processing is not affected by asp3, it sounds reasonable to think that asp3 depletion could have some effect on secreted proteins although their processing remains independent of ASP3. Is there any assay that could determine whether MIC8 is still secreted in the absence of ASP3?*

Addressing this point deserves first a clarification: RON4 processing as well as the processing of ALL RONs and the ROPs examined in this study are mediated by ASP3!

As recommended, we have examined the secretion as well as cleavage of MIC8 in parasites depleted in ASP3, and observed no obvious defects. This data has been added to Figure 3—figure supplement 1.

We favor more the impact of TgASP3 on a yet unidentified factor contributing to rhoptry discharge but that will await further investigations.

*The authors report the localization of Asp3 and tried to validate its localisation in Golgi and endosomal like compartments. The protein was however not found residing in these compartments (Figure 1). Rather it is expected to localize to post Golgi compartment. Ectopically expressed and the WT Asp3 have been reported to localise in this sub-compartment by Shea et al., 2007, which authors have acknowledged. In this case for further proof, experiment using brefeldin A (PMID: 1629235, PMID: 11532123) is advisable. By inhibition of trans Golgi transport, the substrates of Asp3 are expected to accumulate in the post Golgi region. And, if authors can use any known marker of post Golgi compartments, the localisation of Asp3 will be more clear.*

Brefeldin A inhibits protein transport from the endoplasmic reticulum (ER) to the Golgi so it would not really help clarifying the localization of ASP3. Also long term treatment of parasites with Brefeldin A not only collapses the early secretory pathway but also deforms the parasites to a round shape that makes IFA difficult to interpret.

Instead and more informatively, the sensitivity of ASP3 mediated processing to Brefeldin A has actually been reported before on ROP1, an ASP3 substrate(Soldati et al., 1998). In this study, metabolic labeling coupled to pulse-chase experiments established that ROP1 processing was Brefeldin A sensitive and thus occurring post ER. Moreover and importantly the same pulse-chase experiments were also performed in combination with 15°C and 20°C temperature blocks known to prevent transport in a pre-Golgi compartment (like Brefeldin A) and in the trans-Golgi network, respectively. These functional data supported unambiguously a post-Golgi compartment for the maturase of ROP1.

Concordantly, ASP3 localizes in a post-Golgi compartment as shown by IFA with the markers GRASP (cis-Golgi), DrpB (post-Golgi), and proM2AP (ELC). Our attempts to localize TgASP3 by iEM, have disappointingly failed so far.